# On the Predictive Power of Representation Dispersion in Language Models

**Yanhong Li**[1]    **Ming Li**[2]    **Karen Livescu**[3]    **Jiawei Zhou**[4]
[1]Allen Institute for AI    [2]University of Maryland    [3]Toyota Technological Institute at Chicago
[4]Stony Brook University
yanhongl@allenai.org    jiawei.zhou.1@stonybrook.edu

## Abstract

We show that a language model's ability to predict text is tightly linked to the *breadth* of its embedding space: models that spread their contextual representations more widely tend to achieve lower perplexity. Concretely, we find that *representation dispersion*—the average pairwise cosine distance among hidden vectors—strongly and negatively correlates with perplexity across diverse model families (LLaMA, Qwen, and others) and domains (Wikipedia, news, scientific abstracts). Beyond illustrating this link, we show how dispersion can be leveraged for a range of practical tasks—without requiring labeled data. First, measuring dispersion on unlabeled text allows us to rank examples by difficulty and identify hard slices in new domains, offering a data-efficient tool for screening and prioritizing models before full evaluation. Next, we find that identifying layers with higher dispersion pinpoints the best representations for retrieval-based methods such as $k$NN-LM, bypassing exhaustive layer-by-layer searches. Finally, we integrate a simple "push-away" objective into training, which increases dispersion in both single-domain and cross-domain scenarios and directly improves perplexity in each.[1]

## 1 Introduction

Large language models can perform remarkably well on tasks ranging from text completion to code generation. Yet their *embedding geometry* often exhibits signs of anisotropy or rank collapse, whereby hidden states lie in a narrow cone or occupy a low-dimensional subspace (Ethayarajh, 2019; Gao et al., 2019; Li et al., 2020). Although this geometry has been posited to limit expressive power, *how* precisely it connects to auto-regressive text generation remains less clear.

In this paper, we present empirical evidence that a model's ability to predict text is tightly linked to the *breadth* of its embedding space. Intuitively, as illustrated in Figure 1, weaker models compress contexts into tight clusters, whereas stronger models separate these contexts —*even semantically similar ones*—more broadly. This broader geometry yields clearer distinctions in the latent space, enabling sharper (lower-entropy) next-token predictions. Concretely, we quantify **representation dispersion** at *any chosen layer* as the average pairwise cosine distance of its hidden vectors. Unless specified otherwise, our empirical sections use the final layer, and we show that higher dispersion consistently predicts *lower perplexity* (Figure 2).

Beyond revealing this fundamental link, representation dispersion offers *practical* benefits:

- **Label-free diagnostics.** Dispersion measured on *unlabeled* text monotonically tracks correctness for a fixed model and dataset, enabling label-free *ranking* of examples by difficulty and discovery of hard slices (§3.1).
- **Model selection.** Among multiple pretrained or fine-tuned variants within the same model family, larger dispersion provides a coarse, inexpensive signal that helps identify clearly under-adapted checkpoints and prioritize promising ones for full evaluation (§3.2).
- **Layer selection for retrieval augmentation.** Although the first two applications exploit the final hidden state, dispersion is equally informative inside the network. In $k$NN-LM, choosing

---

[1]Code is available at https://github.com/yanhong-lbh/rep_dispersion.

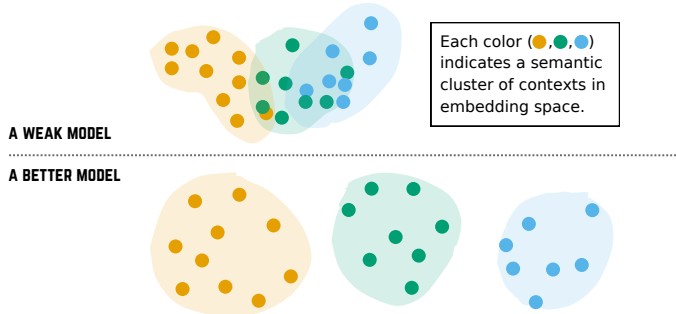

Figure 1: **Representation geometry in a weak model vs. a better model.** In a weak model (top), final-layer embeddings for similar contexts are compressed into tight clusters, limiting discriminative power. In a better model (bottom), embeddings are widely dispersed—even within semantically related clusters—leading to more confident next-token predictions.

Figure 2: **An Illustrative Empirical Trend.** Across models/datasets, higher dispersion correlates with lower perplexity.

the hidden layer with the highest dispersion yields the best perplexity, providing an unsupervised shortcut to sub-layer selection (§3.3).

- **A training signal.** Encouraging higher dispersion through an auxiliary "push-away" loss directly improves perplexity for both single-domain and cross-domain scenarios (§3.4).

Overall, we show that a model's embedding geometry—captured by the simple statistic of *average pairwise distance*—serves as a robust indicator of predictive quality. By quantifying and encouraging this broader geometry, we gain both conceptual insight and practical benefits, and we hope this perspective fosters new avenues for improving model robustness and interpretability.

## 2 EMPIRICAL ANALYSIS OF REPRESENTATION GEOMETRY

We begin by describing how we measure representation geometry and perplexity (§2.1). We then present three key global observations that characterize how embedding dispersion evolves with perplexity, across layers, and under fine-tuning (§2.2). Finally, we zoom in on semantic clusters (§2.3) to examine how dispersion behaves among closely related contexts.

### 2.1 MEASUREMENT SETUP

**Contextual Representations.** Following standard autoregressive conventions, let a language model with parameters $\theta$ assign probability to a token sequence $(x_1, x_2, \ldots, x_N)$ via $p_\theta(x_1, x_2, \ldots, x_N) = \prod_{n=1}^{N} p_\theta(x_n \mid x_{<n})$, where $x_{<n} = (x_1, \ldots, x_{n-1})$. In contemporary Transformer-based models, each partial sequence $x_{<n}$ is mapped to an internal *context vector* $\mathbf{h}_n \in \mathbb{R}^d$ by a function $f_\theta(\cdot)$. The next-token distribution is then given by $p_\theta(x_n \mid x_{<n}) = \text{softmax}(W_o \mathbf{h}_n)$, where $W_o \in \mathbb{R}^{|V| \times d}$ projects the representation $\mathbf{h}_n$ into the vocabulary space $V$. For measuring *representation dispersion*, we focus on a chosen final-layer vector (e.g., $\mathbf{h}_N$ for the full sequence), by default, as the representation of each text sample.

**Measuring Representation Dispersion.** To measure how well the model separates text samples in its embedding space, we first choose a particular layer or sub-layer whose representation we wish to examine (e.g., the final hidden state, the output after the final attention block, or the second-to-last block). Concretely, we sample $N$ text segments from a dataset and pass each segment through the model to extract the corresponding representation $\mathbf{E}_i \in \mathbb{R}^d$ from the chosen sublayer, for $i = 1, \ldots, N$. We then compute the **average pairwise cosine distance** of these representations:

$$\overline{D} = \frac{1}{\binom{N}{2}} \sum_{1 \le i < j \le N} \left[ 1 - \frac{\mathbf{E}_i \cdot \mathbf{E}_j}{\|\mathbf{E}_i\| \, \|\mathbf{E}_j\|} \right]. \tag{1}$$

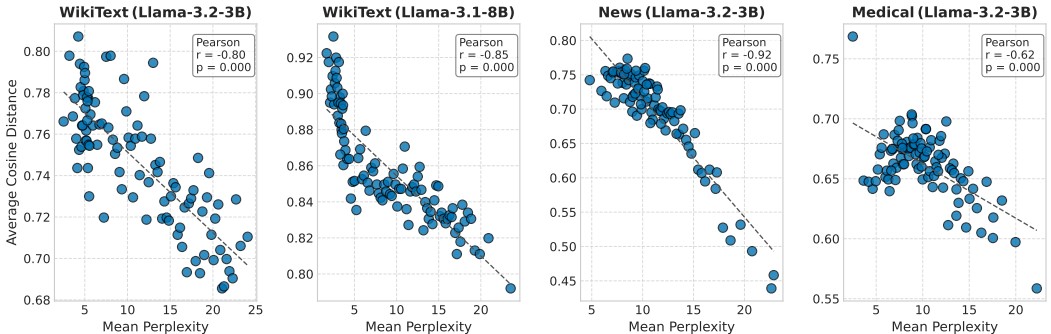

Figure 3: Sequence-level perplexity vs. embedding dispersion across different domains and model sizes. Each point represents a bin of text segments with the x-axis showing mean sequence-level perplexity and the y-axis showing average pairwise cosine distance of final-layer embeddings.

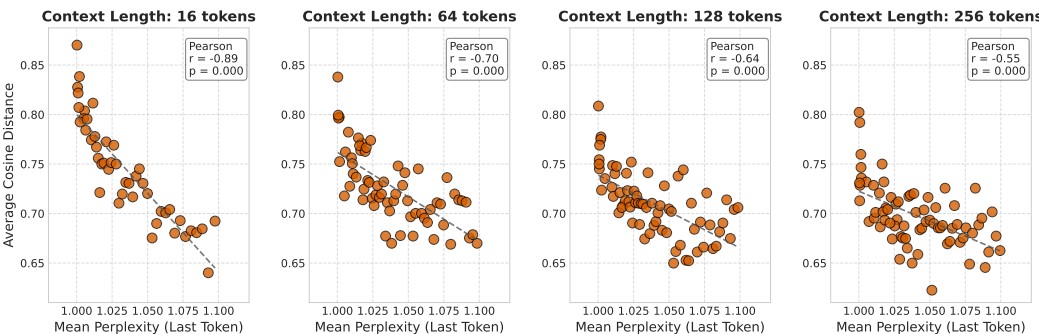

Figure 4: Last-token perplexity vs. embedding dispersion at varying context lengths using LLaMA-3.2-1B. Each point represents a bin of text segments with the x-axis showing mean last-token perplexity and the y-axis showing average pairwise distance.

This quantity reflects how "spread out" the embeddings are, with higher values indicating greater separation among representations.

## 2.2 GLOBAL OBSERVATIONS ON REPRESENTATION DISPERSION

In this section, we examine how the model's representation space behaves across a broad sample of text segments, highlighting how perplexity, layer depth, and fine-tuning each affect embedding dispersion.

**(1) Perplexity vs. Representation Dispersion.** Our first finding connects a model's *sequence-level perplexity* to how spread out its contextual embeddings are.[2] We randomly select 100,000 text segments of 512 tokens, compute their perplexities, and also measure the **average pairwise distance** of their final-layer embeddings. We sort by perplexity and group segments into bins, and for each bin recording its mean perplexity and mean pairwise distance.

Figure 3 reveals a strong **negative correlation**: Segments with *lower* perplexity have *more dispersed* embeddings, whereas those with *higher* perplexity show *more compressed* embeddings. This trend appears across multiple model families (Llama, Phi, Mistral, Qwen) and diverse text domains (e.g.

---

[2]We define the sequence-level perplexity of a text segment $x_{1:L}$ of length $L$ as:

$$\text{ppl}(x_{1:L}) = \exp\left(-\frac{1}{L}\sum_{t=1}^{L}\log p_\theta(x_t \mid x_{<t})\right),$$

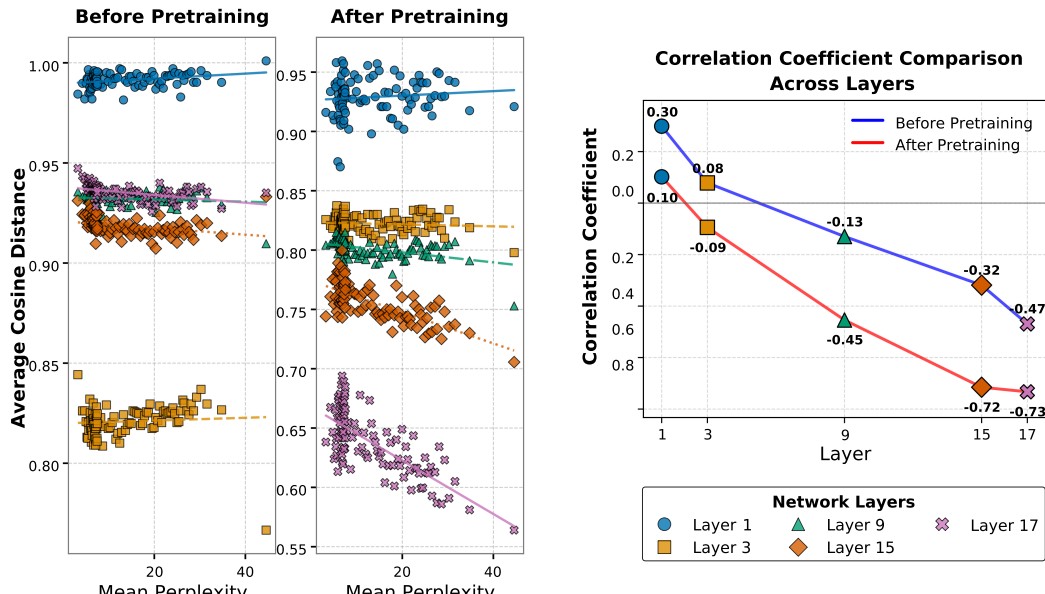

Figure 5: Layer-wise embedding separation in LLaMA-3.2-1B, shown across different perplexity bins (left panel) and the corresponding correlation coefficients (right panel). Note that the negative correlation between perplexity and embedding separation becomes more pronounced as we move to deeper layers (see right panel).

Wikitext-103, CNN Daily News, PubMed summarization). Full visualizations are provided in Section A.1.3.

We also verify that the relationship holds at a finer granularity by focusing on *last-token perplexity*, shown in Figure 4. Across context lengths of 16, 64, 128, and 256 tokens, we observe the same negative correlation trend. Intuitively, this negative correlation appears because contexts with more confident next-token predictions (low perplexity) end up pushed into more distinct regions of the embedding space, whereas harder-to-predict contexts remain more compressed.

**(2) Layer-Wise Patterns.** We next assess how this relationship unfolds across layers. Collecting embeddings from multiple intermediate layers (e.g. Layers 1, 3, 9, 13, 17) and replicating the above procedure, we find that *the negative correlation strengthens in deeper layers* (as illustrated in the right panel of Figure 5). Early layers do not show a clear correlation, likely because they capture lower-level lexical features rather than global predictive cues. Deeper layers exhibit *more pronounced* embedding distance differences between easier-to-predict and harder-to-predict samples. Figure 5 illustrates this layer-wise progression for a representative LLaMA-3.2-1B model. Additionally, comparing models before and after pretraining indicates that the negative correlation emerges primarily after the model has been trained to predict tokens, pointing to a learned representational structure.

**(3) Fine-Tuning Effects.** Finally, we examine how *fine-tuning* reshapes the embedding space. We select the same model (LLaMA-3.2-1B) and apply either parameter-efficient (*LoRA* (Hu et al., 2021)) or *full-parameter* fine-tuning on WikiText-103. Compared to the pre-trained model, both approaches *increase* the average embedding separation on WikiText-103 (Figure 6). Full fine-tuning exerts a stronger effect, pushing text samples further apart overall; LoRA, which adapts only low-rank modules, effects smaller but still notable changes. The choice of fine-tuning thus influences how discriminative (i.e. "spread out") the embeddings become in a specialized domain.

## 2.3 DISPERSION WITHIN SEMANTIC CLUSTERS: A FINER-GRAINED ANALYSIS

A natural question is whether better models merely push *semantically dissimilar* contexts farther apart, or also increase distances among contexts that are *semantically close*. If the latter did not

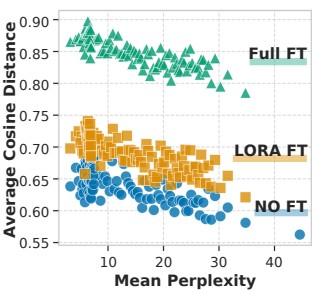

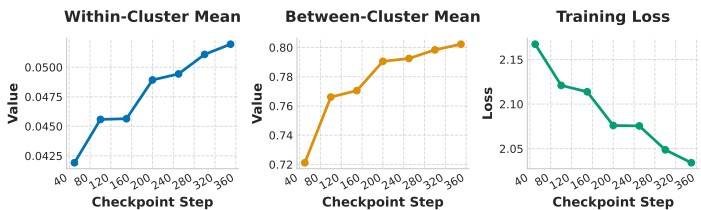

Figure 7: **Clustering metrics and training loss across training.** We form 500 clusters of WikiText-103 contexts (each cluster contains 100 semantically similar contexts that share the same 10-gram continuation). During model training, both the *within-cluster* distance (left) and *between-cluster* distance (middle) consistently grow, while training loss (right) falls.

Figure 6: Effect of fine-Tuning on embedding separation.

happen, then we could imagine a scenario where highly related examples remain tightly clustered, but unrelated examples spread out, thereby inflating overall average distances. We therefore directly measure dispersion among *clusters* of highly similar contexts to see whether their internal geometry also expands over training.

**Setup & Metrics.** We collect 10-grams from the WikiText-103 training set that appear at least 100 times. For each such 10-gram, we retrieve 100 occurrences, each accompanied by a 100-token left context, and treat these contexts as a semantically similar *cluster*. (Manual inspection confirmed that these context sets do indeed relate to the same event, entity, or theme.) We construct a total of 500 such clusters. We then track the contextual embeddings produced by a LLaMA-3.2-1B model at various checkpoints during training. Within each cluster, we compute the *within-cluster distance* as the average (cosine) distance from each embedding to the centroid of that cluster. Likewise, we define the *between-cluster distance* as the average (cosine) distance among the *centroids* of all clusters. Thus, the latter measures how far clusters are from each other in the embedding space, while the former measures how tightly each cluster is packed internally.

**Results.** Figure 7 shows that, as training proceeds and the loss decreases, both the average *within-cluster* distance and *between-cluster* distance increase. Thus, the trend toward higher overall dispersion is not driven solely by pushing apart highly dissimilar contexts. Even very similar contexts— which all share the same 10-gram continuation—become more spread out in the latent space as the model learns, which indicates the model can effectively distinguish them despite the similarities. This in-cluster expansion reinforces our hypothesis that better-performing models tend to produce broader embedding geometries in *all* regions of the latent space, not just pushing away dissimilar examples.

## 3 APPLICATIONS

### 3.1 RANKING EXAMPLE HARDNESS WITHOUT LABELED DATA

In many real-world scenarios, one receives a large *unlabeled* query set and must decide, *before* committing annotation or compute resources, (1) whether an off-the-shelf model will be sufficiently accurate and (2) which specific examples it is likely to get wrong. A reliable *label-free* indicator would enable rapid model validation, automatic justification of *easy* versus *hard* instances, and targeted continued pre-training on precisely those queries that the model currently struggles with. Because higher representation dispersion tracks lower perplexity (§2), we hypothesize that dispersion can serve as such an indicator: if high dispersion coincides with correct predictions, then simply measuring distances allows us to rank examples by expected difficulty and flag likely errors without ever looking at ground-truth labels.

**Experimental protocol.** To test this idea, we design a controlled experiment that varies the *fraction of correct answers* while keeping the input distribution fixed: (1) Collect a pool of question-answer pairs for a given dataset-model pair. (2) Partition the pool into *correct* and *incorrect* subsets by comparing the model's answer with the ground truth. (3) For each desired accuracy level (0%–100%

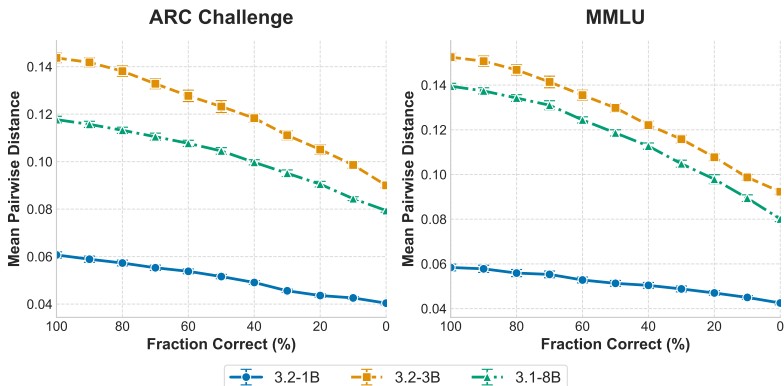

Figure 8: **Embedding distance vs. fraction correct.** Each point corresponds to a fixed mixture of correct/incorrect model predictions (x-axis). Error bars denote standard error over 10 random seeds. Higher accuracy consistently aligns with larger mean pairwise distance. Similar trends are found in multilingual MMLU, HellaSwag, IFEval, and QuAIL. Details in Section B.1.1 and Section B.1.4.

in 10% increments), sample 100 queries that contain the requisite mix of correct and incorrect cases.(4) Extract the final-layer embeddings of these queries and compute their mean pairwise cosine distance, averaging over 10 random seeds.

**Results.**   Figure 8 plots mean pairwise distance against the fraction of correct predictions for several LLaMA variants on ARC-CHALLENGE and MMLU. Across all models and datasets, *accuracy rises monotonically with dispersion*: slices that the model answers correctly exhibit markedly broader geometry than those it answers incorrectly. Practically, a practitioner could therefore *sort* an unlabeled dataset by dispersion, inspect only the low-dispersion tail to uncover failure modes, or focus continued training on those "hard" queries. Taken together, these findings establish representation dispersion as a powerful, zero-label indicator of *relative hardness* and a principled tool for slice discovery and targeted data augmentation, rather than a fully calibrated predictor of absolute accuracy.

## 3.2   REPRESENTATION DISPERSION FOR MODEL SELECTION

In practice, researchers and practitioners are often faced with choosing among numerous model variants—ranging from different instruction-tuned checkpoints to parameter-efficient adaptations or distilled models—while only having limited labeled data in the target domain. Exhaustively evaluating every checkpoint is typically prohibitively expensive, both in terms of computation and annotation resources. The tight link between representation dispersion and predictive accuracy established in §2 offers an attractive alternative: by simply measuring how broadly a model separates key tokens in its embedding space, one can obtain a geometric score that rapidly screens checkpoints *within a fixed model family and tokenizer*.

**Setup.**   To operationalize this idea, we focus on task-relevant tokens that carry significant signal in the domain of interest, such as digits for mathematical reasoning, Python keywords for code generation, or legal terms for contract analysis. Let $\mathcal{T} \subset V$ denote a small set of such domain-specific tokens, and let $\overline{\mathcal{T}}$ denote a reference set of common, everyday language tokens. We use the model's original output token embeddings—that is, the rows of the output projection matrix—as provided after loading the model weights. This requires no forward passes or input data; all computations are performed directly on these pre-trained embeddings. We compute average pairwise distances—either cosine or Euclidean—among the embeddings of tokens within $\mathcal{T}$, within $\overline{\mathcal{T}}$, and between $\mathcal{T}$ and $\overline{\mathcal{T}}$.

Motivated by our empirical findings, we propose a single "dispersion gap" metric that succinctly captures both the distinctiveness of the domain-relevant tokens and their separation from generic language. Specifically, we define

$$\mathcal{G} = \text{within}(\mathcal{T}) + \text{between}(\mathcal{T}, \overline{\mathcal{T}}),$$

where within$(\mathcal{T})$ denotes the mean pairwise distance among the domain tokens, and between$(\mathcal{T}, \overline{\mathcal{T}})$ is the mean distance between domain and reference tokens. Larger values of $\mathcal{G}$ indicate that the model both differentiates between domain-critical tokens and separates them from everyday vocabulary—a geometric pattern that, as Tables 9–12 show, strongly correlates with higher task accuracy.

This approach is computationally efficient and entirely label-free: evaluating $\mathcal{G}$ requires only reading the model's output embedding matrix and performing basic matrix operations on CPU, without any forward passes or GPU computation. In our experiments, ranking models by their dispersion gap within a given family and parameter scale consistently elevates the best-performing models in domains such as math and code, yielding gains of up to 40 accuracy points over clearly under-adapted variants. We therefore view the dispersion gap as a coarse but effective pre-filter: it reliably flags models that have not yet learned the relevant domain geometry, after which more expensive task-specific evaluation can be used to break ties among the remaining strong candidates.

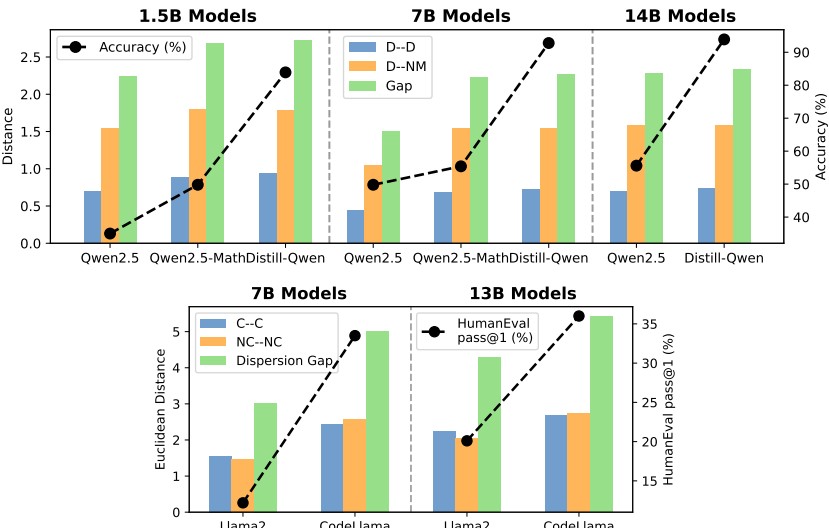

Figure 9: **Upper: Qwen on MATH.** Blue bars show $D–D$ (digit–digit) distances, orange bars show $D–NM$ (digit–non-math) distances, and green bars show their sum—the *Dispersion Gap*. The black dashed line reports task accuracy (%). **Lower: Llama/CodeLlama on HUMANEVAL.** Blue bars show $C–C$ (code-keyword) distances, orange bars show $NC–NC$ (non-code) distances, and green bars again give the *Dispersion Gap*. The black dashed line reports PASS@1 (%). Bar heights therefore convey the three dispersion statistics, while the line traces model performance, mirroring the figure legends. A full table of the underlying values is provided in Section B.2.

**Results.** Figure 9 clearly demonstrates how the dispersion gap $\mathcal{G}$ aligns with downstream performance. In the *upper* panel (Qwen on MATH), accuracy rises monotonically with the green "Gap" bars both *within* each parameter scale (e.g. 1.5 B $\rightarrow$ 7 B $\rightarrow$ 14 B) and *across* fine-tuned variants (e.g. QWEN2.5, QWEN2.5-MATH, DISTILL-QWEN). Spearman correlations between $\mathcal{G}$ and accuracy exceed 0.95, and the dispersion gap perfectly ranks all nine checkpoints without a single mis-ordering. The *lower* panel (Llama/CodeLlama on HUMANEVAL) shows the same pattern: CODELLAMA consistently exhibits both a larger gap and a higher PASS@1 rate than its LLAMA2 counterpart at *both* 7 B and 13 B scales, while the increase from 7 B to 13 B again boosts both metrics in lock-step. To confirm this pattern is robust beyond static model comparisons, Section B.2.2 extends our analysis to the training trajectory of Olmo-7B, showing that dispersion tracks performance improvements across 30 intermediate checkpoints with correlations exceeding 0.90.

## 3.3 LAYER SELECTION FOR κNN-LM

Many retrieval-augmented language models build a datastore by using one of the sublayers in the final transformer block as the vector key (Khandelwal et al., 2020; He et al., 2021; Alon et al., 2022; Li et al., 2024). A central question in these methods is which internal representation of the transformer

| Model | N=10 | | N=50 | | N=100 | |
|---|---|---|---|---|---|---|
| | Attn | FFN | Attn | FFN | Attn | FFN |
| DISTILGPT2 | 0.83±0.02 | 0.33±0.05 | 0.83±0.01 | 0.30±0.02 | 0.82±0.01 | 0.30±0.01 |
| GPT2-SMALL | 0.83±0.02 | 0.24±0.06 | 0.83±0.00 | 0.24±0.03 | 0.83±0.00 | 0.24±0.01 |
| GPT2-MEDIUM | 0.66±0.03 | 0.19±0.02 | 0.67±0.01 | 0.21±0.02 | 0.67±0.01 | 0.21±0.01 |
| GPT2-LARGE | 0.80±0.04 | 0.68±0.06 | 0.79±0.01 | 0.66±0.02 | 0.80±0.01 | 0.68±0.03 |

Table 1: Representation dispersion (average pairwise cosine distance) of the final block's attention and feed-forward sub-layers for varying numbers of randomly sampled chunks $N$. Higher values indicate more widely separated contextual embeddings.

to use as the "datastore key." In this section, we focus on $k$NN-LM, which augments a standard language model with a key-value memory of training examples. Specifically, at inference time, it retrieves the nearest neighbors of the current hidden state from that memory and interpolates their next-token distribution with the base LM's distribution, often achieving lower perplexity on rare or out-of-distribution tokens. Recent evidence (Xu et al., 2023) suggests that taking the attention-layer output (instead of the feed-forward layer output) often improves $k$NN-LM perplexity, but pinpointing the best layer can require expensive end-to-end trials. We show that measuring how widely each layer "disperses" its text embeddings (via average pairwise cosine distance) provides a lightweight, *unsupervised* way to identify a layer likely to yield strong $k$NN-LM performance—without running the full interpolation pipeline.

**Background.** Consider the standard Transformer block as used by GPT-2. Let $\mathbf{h}^{(l-1)} \in \mathbb{R}^{n \times d}$ denote the hidden states from block $l - 1$. In block $l$, the hidden states first pass through a multi-headed attention (MHA) sub-layer and residual connection: $\widehat{\mathbf{h}}^{(l)} = \mathbf{h}^{(l-1)} + \text{Dropout}\Big(\text{MHA}\big(\text{LayerNorm}(\mathbf{h}^{(l-1)})\big)\Big)$. We then apply another residual connection around a position-wise feed-forward network (FFN): $\mathbf{h}^{(l)} = \widehat{\mathbf{h}}^{(l)} + \text{Dropout}\Big(\text{FFN}\big(\text{LayerNorm}(\widehat{\mathbf{h}}^{(l)})\big)\Big)$. Thus, each block produces two intermediate sub-layer outputs: $\mathbf{h}_{\text{att}}^{(l)} = \widehat{\mathbf{h}}^{(l)}$ and $\mathbf{h}_{\text{ffn}}^{(l)} = \mathbf{h}^{(l)}$. Following Xu et al. (2023), we compare using $\mathbf{h}_{\text{att}}^{(L)}$ versus $\mathbf{h}_{\text{ffn}}^{(L)}$ from the *final* block $L$ as the contextual key for $k$NN-LM.

**Experimental Setup.** We consider four members of the GPT-2 family—DISTILGPT2 (82 M parameters), GPT2-SMALL (117 M), GPT2-MEDIUM (345 M), and GPT2-LARGE (774 M)—all trained with the same tokenizer and a context length of 1 024. For each checkpoint we draw $N \in \{10, 50, 100\}$ non-overlapping 512-token *chunks* randomly from the WIKITEXT-103 validation split. Sampling is repeated 10 times with different random seeds, and the identical chunk indices are fed to both sub-layers so that any dispersion difference cannot be attributed to input variance. For every chunk we extract the *last-token* hidden state produced by the final Transformer block's (i) attention sub-layer output $\mathbf{h}_{\text{att}}^{(L)}$ and (ii) feed-forward sub-layer output $\mathbf{h}_{\text{ffn}}^{(L)}$. Given the resulting set of $N$ vectors $\{\mathbf{h}_i\}_{i=1}^N$, we measure their *representation dispersion* as the average pairwise cosine distance. We report the mean and standard deviation of dispersion across the 10 random repeats for every $\langle$model, $N$, sub-layer$\rangle$ triple.

**Results.** Table 1 shows two practitioner-relevant patterns. First, the hidden states taken *after* the attention sub-layer are always more widely spread than those taken after the feed-forward sub-layer (e.g., 0.8045 vs. 0.6831 for GPT2-LARGE, 0.6593 vs. 0.1865 for GPT2-MEDIUM), making $\mathbf{h}_{\text{att}}^{(L)}$ the natural choice of key for $k$NN-LM. Second, dispersion can be estimated with striking efficiency: moving from 10 to 50 or 100 input chunks alters the mean by at most 1.5 % and never changes the layer ranking, so profiling a model requires only about 5,000 tokens and a few milliseconds.

## 3.4 INCORPORATING REPRESENTATION DIVERGENCE INTO TRAINING

While typical cross-entropy training focuses purely on next-token prediction, several studies suggest that explicitly encouraging embedding separation can improve generalization and robustness in language modeling (Gunel et al., 2021; Jain et al., 2023). Inspired by these findings, we augment the

standard language-modeling loss with an auxiliary objective that *pushes apart* hidden-state vectors, aiming to produce more discriminative representations.

**Setup.** We consider two scenarios: a *single-domain* setting, where we train GPT-2 small on WikiText, and a *cross-domain* setting, where we train on WikiText plus Python code. In both cases, let $\{\mathbf{h}_i\}_{i=1}^B$ be the final-layer hidden-state vectors for all token sequences in a batch (flattened across batch and sequence), where $B$ is the batch size. We normalize each vector to unit length, $\tilde{\mathbf{h}}_i = \mathbf{h}_i / \|\mathbf{h}_i\|$. To encourage wider separation, we compute an average pairwise cosine distance $d$ and then add an auxiliary loss $-d$ to the standard cross-entropy loss, weighted by a hyperparameter $\lambda$.

In the **single-domain** setting, $d_{\text{avg}}$ is defined over all pairs in the same batch: $d_{\text{avg}} = \frac{1}{B(B-1)} \sum_{i \neq j} \left[1 - \tilde{\mathbf{h}}_i \cdot \tilde{\mathbf{h}}_j\right]$. In the **cross-domain** setting, we instead compute $d$ only across pairs drawn from different domains (Wiki vs. code) to push embeddings from each domain further apart: $d = \frac{1}{|A||B|} \sum_{i \in A} \sum_{j \in B} \left[1 - \tilde{\mathbf{h}}_i^{(A)} \cdot \tilde{\mathbf{h}}_j^{(B)}\right]$. The total loss in both settings is $\mathcal{L}_{\text{total}} = \mathcal{L}_{\text{CE}} + \lambda \mathcal{L}_{\text{aux}}$, where $\mathcal{L}_{\text{CE}}$ is the standard next-token cross-entropy and $\mathcal{L}_{\text{aux}} = -d_{\text{avg}}$ (single-domain) or $-d$ (cross-domain), and $\lambda$ controls the strength of the auxiliary "spread-out" loss.. We clarify the relation between this auxiliary term and prior contrastive / repulsive losses in Section B.4.3.

**Results.** Table 2 reports test perplexities for various learning rates and auxiliary loss weights $\lambda$. In the **single-domain** (WikiText) setting, introducing the auxiliary spread-out loss yields a slight decrease in perplexity—typically 1–4 points—relative to the baseline, especially early in training. In the **cross-domain** (WikiText+Code) setting, the auxiliary loss produces a much more pronounced reduction in perplexity for both domains. For all learning rates, models trained with a moderate value of $\lambda$ achieve notably lower perplexity on both WikiText and code, suggesting that explicitly pushing apart representations from distinct domains leads to more specialized and di scriminative features. This demonstrates that our approach is particularly effective when bridging heterogeneous data sources.

| LR | | (a) Single-Domain (WikiText) | | | | (b) Cross-Domain (WikiText+Code) | | | | |
|---|---|---|---|---|---|---|---|---|---|---|
| | $\lambda$ | Step = 500 | | Step = 1000 | | $\lambda$ | Step = 500 | | Step = 1000 | |
| | | Base | +Aux | Base | +Aux | | Wiki | Code | Wiki | Code |
| $10^{-3}$ | 0.0 | 226.1 | – | 111.3 | – | 0.0 | 295.6 | 36.9 | 171.7 | 23.8 |
| | 0.1 | – | 217.5 | – | 108.2 | 0.001 | 270.8 | 34.5 | 158.8 | 22.3 |
| $7 \times 10^{-4}$ | 0.0 | 195.0 | – | 96.7 | – | 0.0 | 304.4 | 35.4 | 175.7 | 22.8 |
| | 0.1 | – | 193.8 | – | 93.6 | 0.01 | 255.2 | 31.9 | 150.2 | 20.8 |
| $5 \times 10^{-4}$ | 0.0 | 166.2 | – | 83.0 | – | 0.0 | 268.5 | 33.7 | 166.9 | 22.1 |
| | 0.1 | – | 165.6 | – | 82.0 | 0.02 | 253.3 | 30.5 | 155.2 | 20.5 |

Table 2: **Auxiliary spread-out loss improves perplexity in both single- and cross-domain settings.** Left: single-domain (WikiText) results; right: cross-domain (WikiText+Code) results. $\lambda$ is the auxiliary loss weight, chosen by validation for each learning rate. We report test-set perplexities at 500 and 1000 steps.

## 4 RELATED WORK

**Geometric Analysis of Embeddings in Language Models.** A growing body of work has examined the geometry of hidden representations in large language models (LLMs). Early studies identified an anisotropy problem, where embeddings collapse into a narrow cone and lose expressiveness at deeper layers (Mu & Viswanath, 2018; Ethayarajh, 2019; Gao et al., 2019; Li et al., 2020; Noci et al., 2022). Recent work uses intrinsic dimension (ID) estimators to trace how representation manifolds evolve across layers (Valeriani et al., 2023), linking geometry to performance through, for example, distinguishing human vs. machine text (Tulchinskii et al., 2023), predicting data compressibility (Cheng et al., 2023), and revealing simplex-like structures for categorical concepts (Park et al., 2025).

The study most closely related to ours is Viswanathan et al. (2025), which also analyzes token-level embedding distributions and observes cosine similarity rises when tokens in the prompt are shuffled. However, their work remains largely descriptive. In contrast, we show how representation dispersion can predict and improve perplexity and downstream accuracy, making geometric insights actionable for model evaluation and selection.

**Mechanistic Interpretability.** Our work offers a geometric perspective on interpretability, complementing research into model mechanisms. This field often reverse-engineers models into interpretable circuits (Elhage et al., 2021; Bereska & Gavves, 2024), identifying specific components like induction heads for in-context learning (Olsson et al., 2022) or analyzing training dynamics through layer-wise gradients (Li et al., 2025b;a). Rather than dissecting individual components, we provide a higher-level, component-agnostic metric by quantifying the dispersion of representations—an abstract but behaviorally meaningful property of the model's internal geometry.

Our method also differs from post-hoc attribution and probing techniques. Attribution methods like Integrated Gradients assign credit to inputs for a given output (Sundararajan et al., 2017; Lundberg & Lee, 2017), while probing trains auxiliary classifiers on hidden states to identify encoded linguistic features (Belinkov, 2022). These approaches typically require labeled data or external probes. In contrast, representational dispersion is a label-free, intrinsic measure; we directly relate the geometry of hidden states to the model's own performance metrics, such as perplexity and downstream accuracy.

## 5 CONCLUSION

In this work, we showed that representation dispersion serves as both a practical diagnostic and training signal for language models. Moving forward, we aim to investigate how representation dispersion interacts with other design choices—such as architectural variations or tokenization strategies—and whether additional regularization signals might further strengthen model robustness and interpretability. We hope these directions will inspire new ways to harness embedding geometry for next-generation language modeling and related tasks.

## STATEMENT ON LLM USAGE

We acknowledge the use of Large Language Models (LLMs) to assist in the preparation of this manuscript. Specifically, LLMs were utilized to improve grammar and clarity, aid in literature discovery, and generate boilerplate code snippets for our experiments and testing scripts. The authors have carefully reviewed and edited all LLM-generated outputs and take full responsibility for the final content and scientific integrity of this work.

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

# Technical Appendices

ORGANIZATION OF CONTENTS

# A   SUPPLEMENTAL MATERIALS FOR REPRESENTATION GEOMETRY ANALYSIS

## A.1   DETAILS REGARDING SEQUENCE-LEVEL PERPLEXITY EXPERIMENTS

### A.1.1   DATASETS AND MODELS

In this section, we provide additional experimental details and visualizations that supplement our main empirical analysis in §2. We study a range of standard language modeling datasets, including the Salesforce/wikitext [3], abisee/cnn_dailymail [4], and ccdv/pubmed-summarization [5], covering text segments in diverse domains.

For the models, our experiments encompass:

- **Llama** families: `meta-llama/Llama-3.2-1B`, `meta-llama/Llama-3.2-3B`, `meta-llama/Llama-3.1-8B`
- **Gemma** families: `google/gemma-2-2b`, `google/gemma-2-9b`
- **Mistral**: `mistralai/Mistral-7B-v0.1`
- **Phi**: `microsoft/phi-2`
- **Qwen** families: `Qwen/Qwen2.5-0.5B`, `Qwen/Qwen2.5-3B`, `Qwen/Qwen2.5-7B`

We use the Hugging Face implementation of the above models. All models are standard decoder-only Transformers, for which we collect final-layer embeddings on randomly selected text segments. In line with Equation 1 of the main paper, we measure average pairwise cosine distance to quantify how "spread out" their representations are.

### A.1.2   PROCEDURE FOR MEAN-PERPLEXITY VS. DISPERSION ANALYSIS

Here, we outline the steps needed to produce a mean-perplexity vs. representation-dispersion plot:

**Step 1:**  Randomly sample 100,000 segments (e.g., 512 tokens each) from the data.

**Step 2:**  For each segment:

    a)  Compute its perplexity over the full sequence.

    b)  Record the final-layer hidden states for later analysis.

**Step 3:**  Sort all segments by their computed perplexity.

**Step 4:**  Group the sorted segments into bins (e.g., 100 segments per bin) and record each bin's mean perplexity.

**Step 5:**  Perform uniform sampling in perplexity space on these bins to ensure coverage of low-, mid-, and high-perplexity regions.

**Step 6:**  For each uniformly sampled bin:

    a)  Retrieve the saved hidden states.

    b)  Calculate pairwise distances (e.g., average cosine distance) among the segment embeddings.

**Step 7:**  Produce the final mapping of mean perplexity to average pairwise distance.

**Uniform Perplexity Sampling.**   Since random sampling of text segments often yields a distribution heavily concentrated around moderate perplexities, we use a uniform sampling scheme to cover both low- and high-perplexity "tails." The pseudocode below highlights the procedure used in **Step 5** (Algorithm 1):

---

[3] http://huggingface.co/datasets/Salesforce/wikitext
[4] https://huggingface.co/datasets/abisee/cnn_dailymail
[5] https://huggingface.co/datasets/ccdv/pubmed-summarization

---

**Algorithm 1** Uniform Perplexity Binning

---

**Require:** A sorted list of $G$ perplexity bins $\{b_1, \ldots, b_G\}$ (with means $m_1 \leq \cdots \leq m_G$)
**Require:** Desired number of bins $K$
**Ensure:** A set of $K$ bins sampled uniformly in perplexity
  1: $m_{\min} \leftarrow m_1$;   $m_{\max} \leftarrow m_G$
  2: Define targets

$$t_k \;=\; m_{\min} + \frac{k-1}{K-1}\left(m_{\max} - m_{\min}\right) \quad \text{for } k = 1, \ldots, K$$

  3: $selected \leftarrow \emptyset$
  4: **for** $k \leftarrow 1$ **to** $K$ **do**
  5:     find $j$ s.t. $m_j$ is closest to $t_k$
  6:     $selected \leftarrow selected \cup \{j\}$
  7: **end for**
  8: **if** $|selected| < K$ **then**
  9:     add extra bins from the sorted list until you have $K$
 10: **end if**
 11: **return** $\{ b_j : j \in selected \}$

---

This ensures we sample across the entire perplexity spectrum, capturing both rare, low-ppl segments and rare, high-ppl segments. With these selected bins in hand, we can then compute the final-layer embeddings and measure representation dispersion to obtain a mean-ppl vs. dispersion plot.

### A.1.3 ADDITIONAL VISUALIZATIONS

Below, we present the full set of perplexity-versus-dispersion plots referenced in §2.2. For each dataset and model, we group 100,000 text segments into perplexity bins and compute their average pairwise representation distances. As described in the main text, we observe a negative correlation between sequence-level perplexity and representation dispersion.

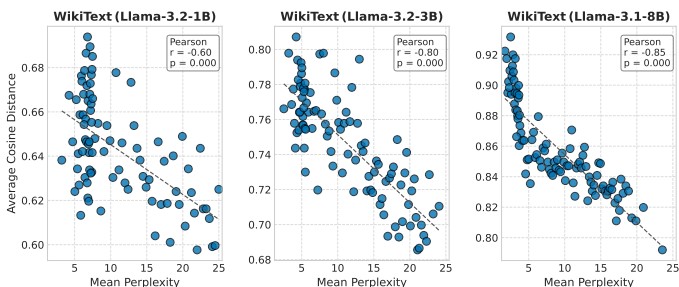

Figure 10: Perplexity vs. Average Pairwise Cosine Distance on Wikitext-103 (Llama family).

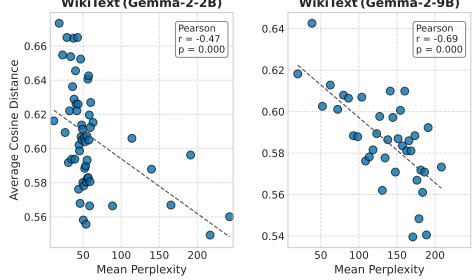

Figure 11: Perplexity vs. Average Pairwise Cosine Distance on Wikitext-103 (Gemma family).

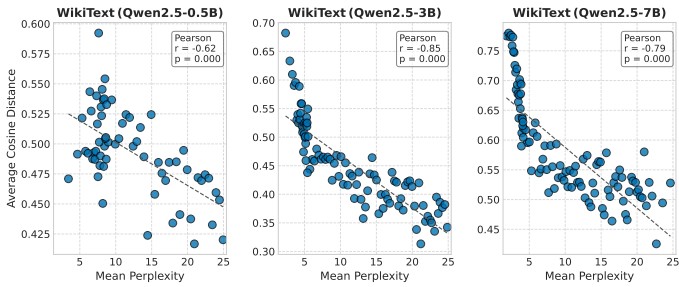

Figure 12: Perplexity vs. Average Pairwise Cosine Distance on Wikitext-103 (Qwen family).

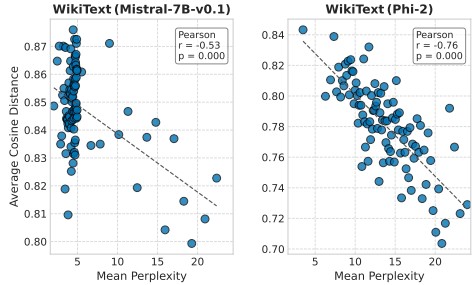

Figure 13: Perplexity vs. Average Pairwise Cosine Distance on Wikitext-103 (Mistral) and Phi.

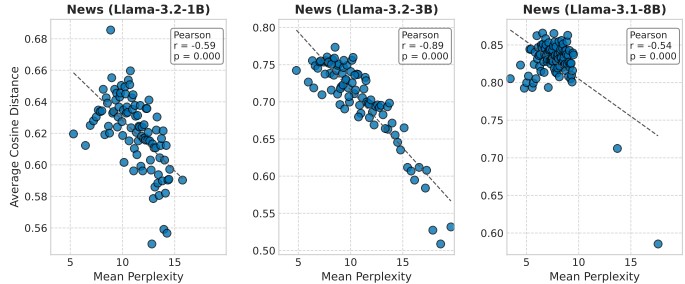

Figure 14: Perplexity vs. Average Pairwise Cosine Distance on CNN DailyMail (Llama family).

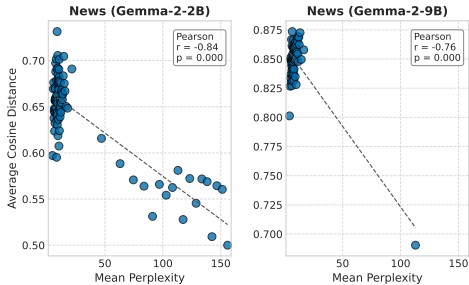

Figure 15: Perplexity vs. Average Pairwise Cosine Distance on CNN DailyMail (Gemma family).

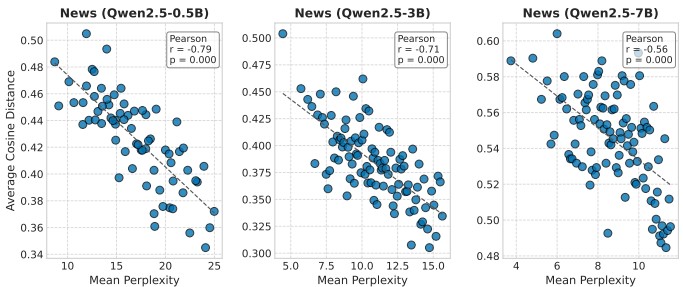

Figure 16: Perplexity vs. Average Pairwise Cosine Distance on CNN DailyMail (Qwen family).

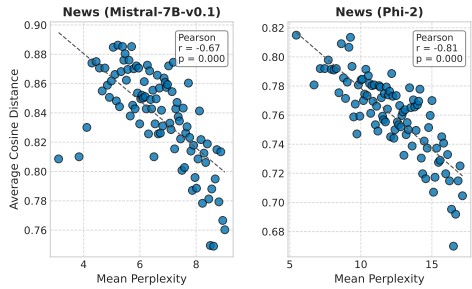

Figure 17: Perplexity vs. Average Pairwise Cosine Distance on CNN DailyMail (Mistral, Phi).

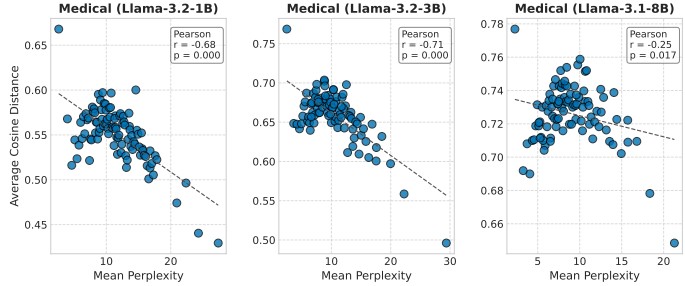

Figure 18: Perplexity vs. Average Pairwise Cosine Distance on PubMed Summarization (Llama family).

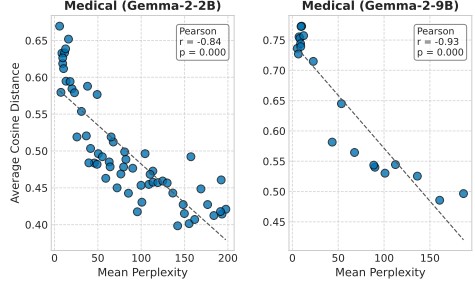

Figure 19: Perplexity vs. Average Pairwise Cosine Distance on PubMed Summarization (Gemma family).

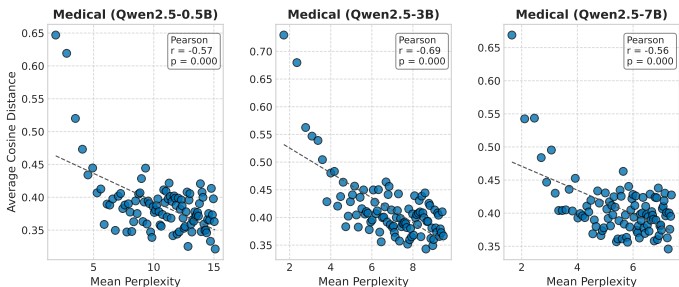

Figure 20: Perplexity vs. Average Pairwise Cosine Distance on PubMed Summarization (Qwen family).

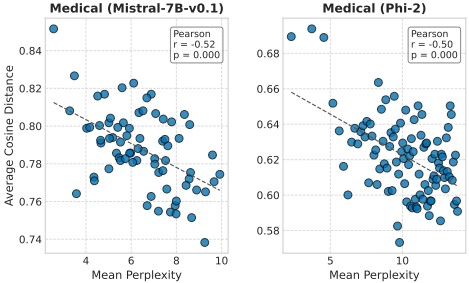

Figure 21: Perplexity vs. Average Pairwise Cosine Distance on PubMed Summarization (Mistral, Phi).

As shown in the figures, the observed *negative correlation* between sequence-level perplexity and representation dispersion holds consistently across:

- Multiple model sizes and architectures (Llama, Gemma, Mistral, Phi, Qwen).
- Multiple data domains (Wikitext-103, CNN DailyMail, PubMed Summarization).

These findings support the main paper's claim that *lower-perplexity* contexts tend to occupy more "spread out" regions in the final-layer embedding space, while *higher-perplexity* (i.e., more challenging) contexts appear more compressed.

## A.2 Details regarding Fine-Tuning Effect Experiments

In §2.2, we examined how fine-tuning influenced representation dispersion. Below are the hyperparameters for the two fine-tuned LLaMA-3.2-1B models used in our experiments. We fine-tuned both checkpoints using the open-source LLaMA-Factory framework [6].

### A.2.1 LoRA Fine-Tuned Model

This model is a fine-tuned version of `meta-llama/Llama-3.2-1B` on the Wikitext-103 dataset. It achieved the following on the evaluation set:

- **Loss:** 2.1764

**Training Hyperparameters.**

- `learning_rate`: 0.0001
- `train_batch_size`: 8
- `eval_batch_size`: 1
- `seed`: 42
- `gradient_accumulation_steps`: 8
- `total_train_batch_size`: 64
- `optimizer`: `adamw_torch` with $\beta_1 = 0.9$, $\beta_2 = 0.999$, $\epsilon = 1 \times 10^{-8}$, no additional arguments
- `lr_scheduler_type`: cosine
- `lr_scheduler_warmup_ratio`: 0.1
- `num_epochs`: 1.0

### A.2.2 Full-Parameter Fine-Tuned Model

This model is also a fine-tuned version of `meta-llama/Llama-3.2-1B` on the Wikitext-103 dataset. It achieved the following on the evaluation set:

- **Loss:** 2.1333

**Training Hyperparameters.**

- `learning_rate`: 1e-05
- `train_batch_size`: 2
- `eval_batch_size`: 1
- `seed`: 42
- `distributed_type`: multi-GPU
- `num_devices`: 2
- `gradient_accumulation_steps`: 16
- `total_train_batch_size`: 64
- `total_eval_batch_size`: 2
- `optimizer`: Adam with $\beta_1 = 0.9$, $\beta_2 = 0.999$, $\epsilon = 1 \times 10^{-8}$
- `lr_scheduler_type`: cosine
- `lr_scheduler_warmup_ratio`: 0.1
- `num_epochs`: 5.0

---

[6]`https://github.com/hiyouga/LLaMA-Factory`

### A.3 DETAILS REGARDING DISPERSION WITHIN SEMANTIC CLUSTERS TRAINING HYPERPARAMETERS

In §2.3, we examined how dispersion evolves within carefully constructed *semantic clusters* of text segments that share the same 10-gram continuation. Below are the training hyperparameters for the model used in this experiment:

- `learning_rate`: 1e-05
- `train_batch_size`: 10
- `eval_batch_size`: 1
- `seed`: 42
- `distributed_type`: multi-GPU
- `num_devices`: 8
- `gradient_accumulation_steps`: 8
- `total_train_batch_size`: 640
- `total_eval_batch_size`: 8
- `optimizer`: ADAMW_TORCH with $\beta_1 = 0.9$, $\beta_2 = 0.999$, $\epsilon = 1 \times 10^{-8}$, no additional arguments
- `lr_scheduler_type`: cosine
- `lr_scheduler_warmup_ratio`: 0.1
- `num_epochs`: 5.0

We used these hyperparameters to train the model from a checkpoint of `meta-llama/Llama-3.2-1B` on WikiText-103, then tracked within-cluster and between-cluster distances of the resulting contextual embeddings at several checkpoints during training. The model is also fine-tuned using the open-source LLaMA-Factory framework.

## A.4 GENERALIZATION TO HELD-OUT DATA

To verify that the correlation between dispersion and perplexity holds on unseen data, we compare the standard distribution (Validation) against a strict Held-Out (Test) split using LLAMA-3.2-1B.

As shown in Figure 22, the negative correlation is consistent across both settings. While the test split (Right) shows higher perplexity overall, the geometric relationship remains: lower perplexity samples are associated with higher embedding dispersion.

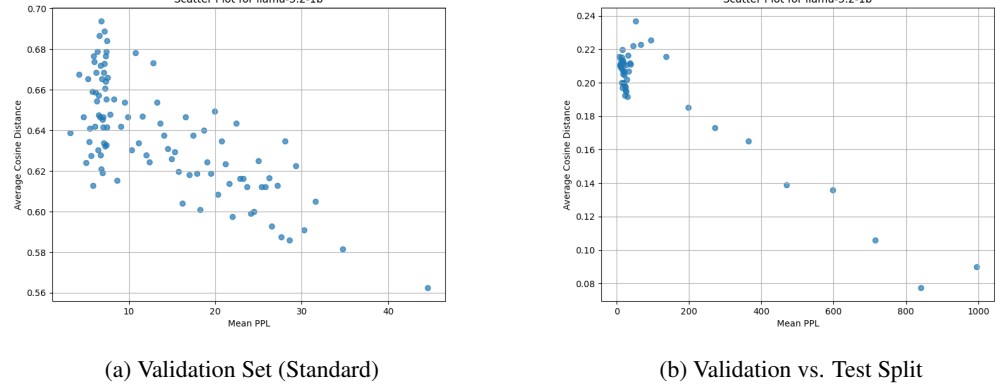

(a) Validation Set (Standard)       (b) Validation vs. Test Split

Figure 22: **Generalization Check (LLAMA-3.2-1B).** Left: The standard correlation on the validation set. Right: The correlation on the held-out test set (orange) vs validation (blue). The negative slope remains robust on unseen data.

## A.5 VENDI SCORE ANALYSIS

To ensure our results are not specific to the Cosine Distance metric, we replicate our analysis using the **Vendi Score** (Friedman & Dieng, 2023). Figure 23 presents a side-by-side comparison of Average Cosine Distance (Left Column) and Vendi Score (Right Column) across three model scales.

In all cases (LLAMA-3.2-1B, 3.2-3B, and 3.1-8B), the Vendi Score follows the exact same trend as Cosine Distance: segments with lower perplexity exhibit higher diversity scores. This confirms that our main metric is a reliable proxy for the intrinsic dimensionality and spread of the representation space.

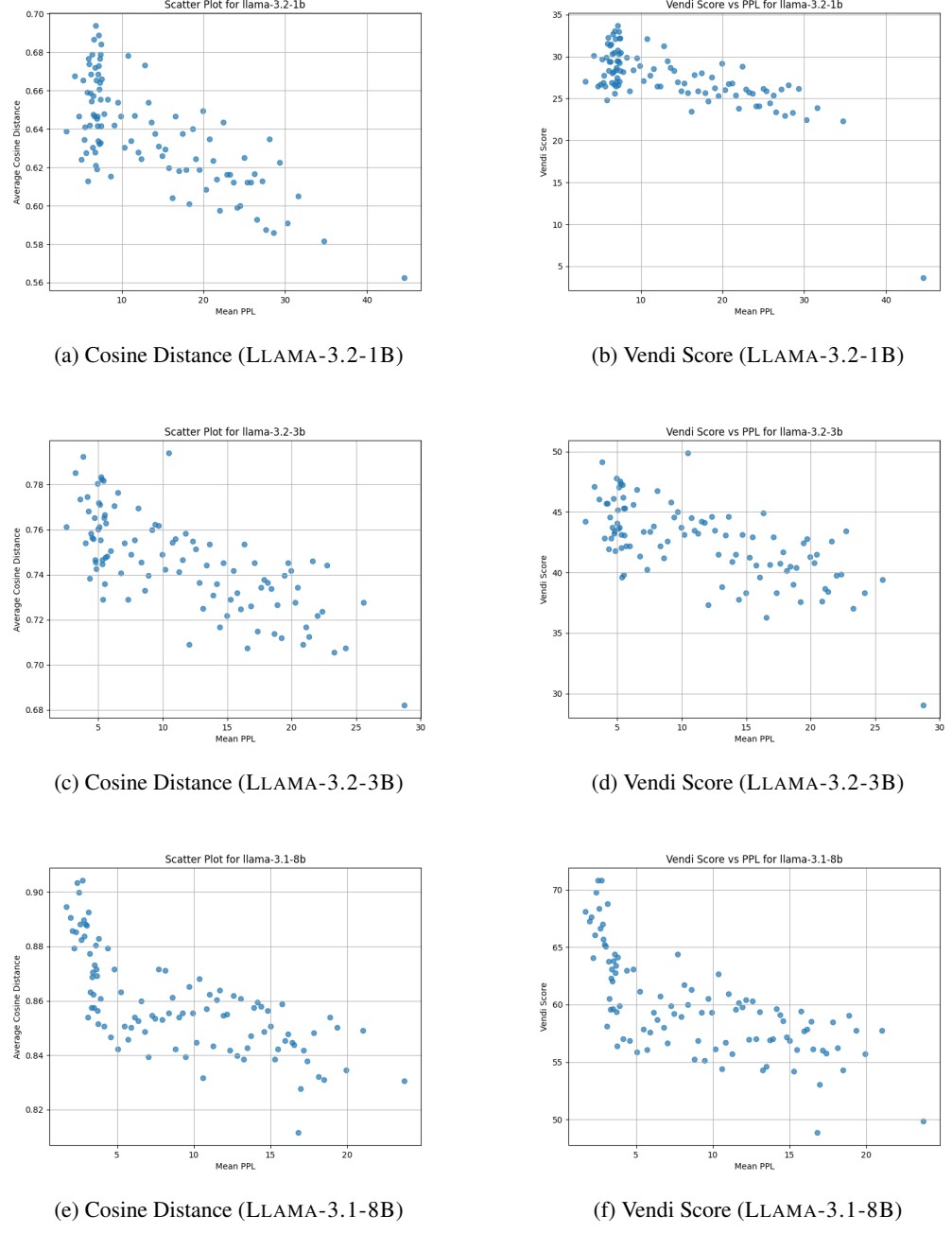

Figure 23: **Metric Robustness across Model Scales.** We compare our primary metric (Average Cosine Distance, left) with the Vendi Score (right) across 1B, 3B, and 8B parameter models. The structural relationship with perplexity is identical across both metrics, validating the robustness of our geometric analysis.

# B SUPPLEMENTAL MATERIALS FOR APPLICATIONS OF REPRESENTATION DISPERSION

## B.1 DETAILS REGARDING RANKING EXAMPLE HARDNESS WITHOUT LABELED DATA

### B.1.1 ADDITIONAL RESULTS

Below we provide extended experimental results following the methodology of §3.1. Each figure contains results for three models: **Llama-3.2-1B-Instruct**, **Llama-3.2-3B-Instruct**, and **Llama-3.1-8B-Instruct**.

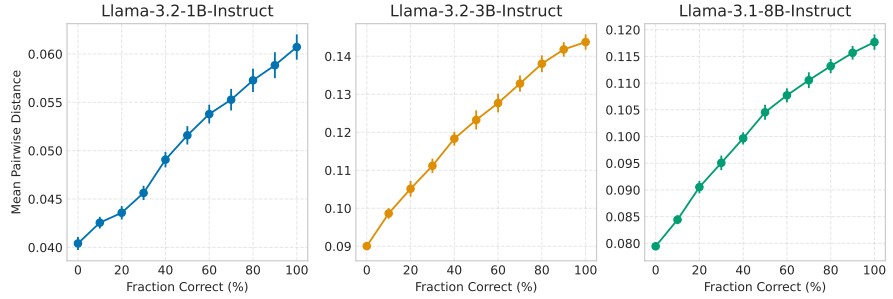

Figure 24: Downstream performance estimation on ARC Challenge (containing results for Llama-3.2-1B-Instruct, Llama-3.2-3B-Instruct, and Llama-3.1-8B-Instruct).

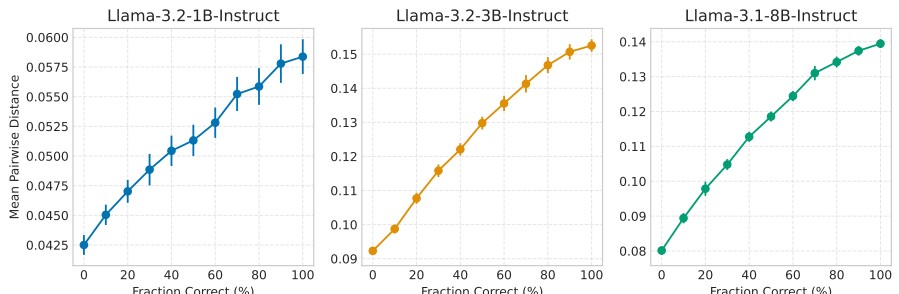

Figure 25: Downstream performance estimation on MMLU (English) (containing results for Llama-3.2-1B-Instruct, Llama-3.2-3B-Instruct, and Llama-3.1-8B-Instruct).

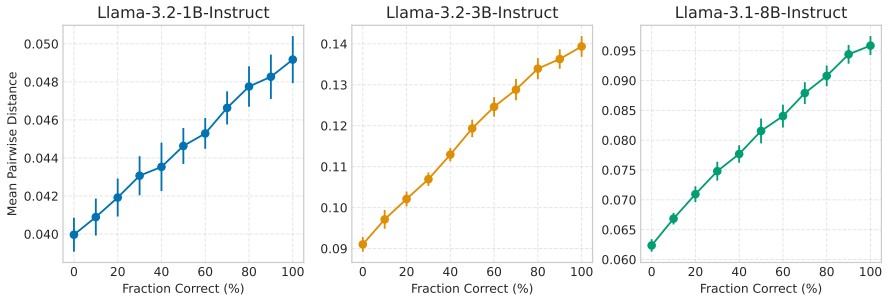

Figure 26: Downstream performance estimation on Multilingual MMLU (German) (containing results for Llama-3.2-1B-Instruct, Llama-3.2-3B-Instruct, and Llama-3.1-8B-Instruct).

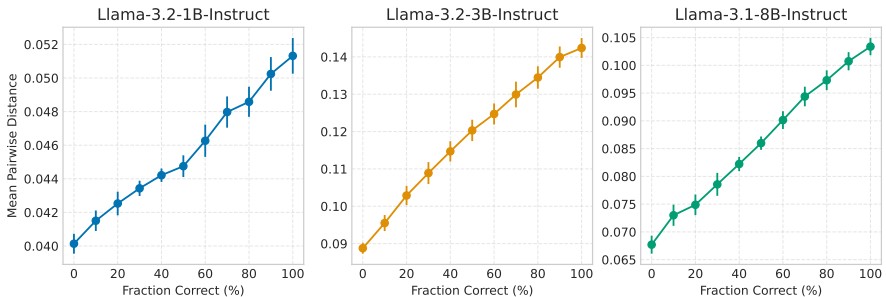

Figure 27: Downstream performance estimation on Multilingual MMLU (Spanish) (containing results for Llama-3.2-1B-Instruct, Llama-3.2-3B-Instruct, and Llama-3.1-8B-Instruct).

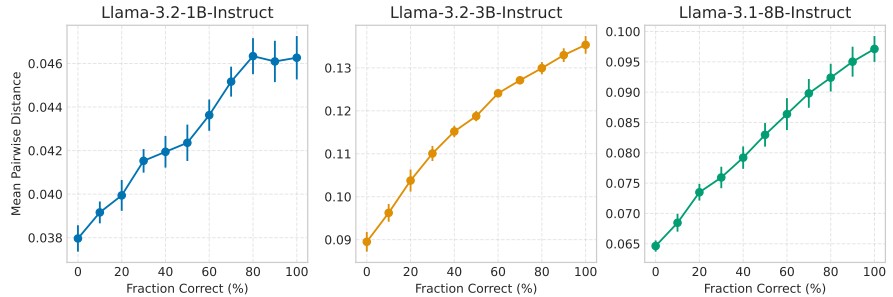

Figure 28: Downstream performance estimation on Multilingual MMLU (French) (containing results for Llama-3.2-1B-Instruct, Llama-3.2-3B-Instruct, and Llama-3.1-8B-Instruct).

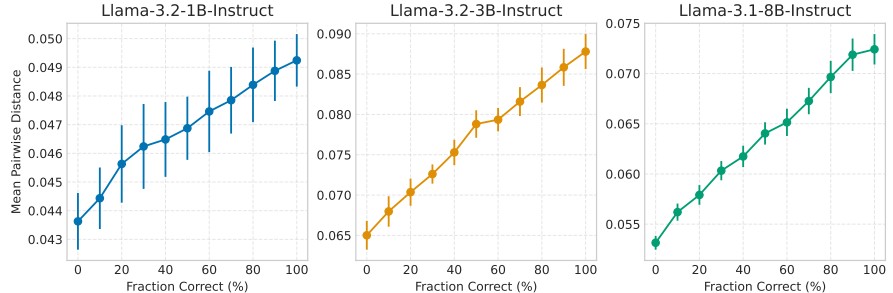

Figure 29: Downstream performance estimation on Multilingual MMLU (Hindi) (containing results for Llama-3.2-1B-Instruct, Llama-3.2-3B-Instruct, and Llama-3.1-8B-Instruct).

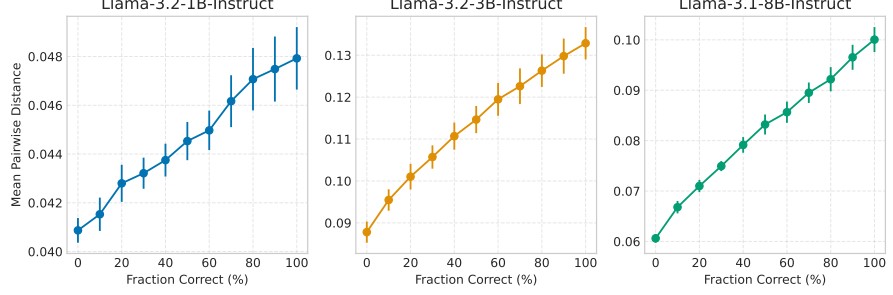

Figure 30: Downstream performance estimation on Multilingual MMLU (Italian) (containing results for Llama-3.2-1B-Instruct, Llama-3.2-3B-Instruct, and Llama-3.1-8B-Instruct).

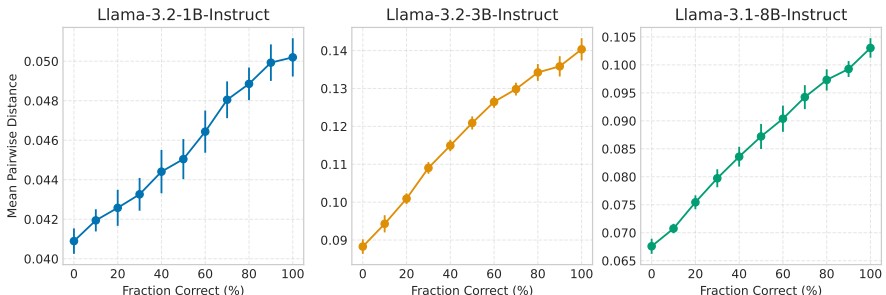

Figure 31: Downstream performance estimation on Multilingual MMLU (Portuguese) (containing results for Llama-3.2-1B-Instruct, Llama-3.2-3B-Instruct, and Llama-3.1-8B-Instruct).

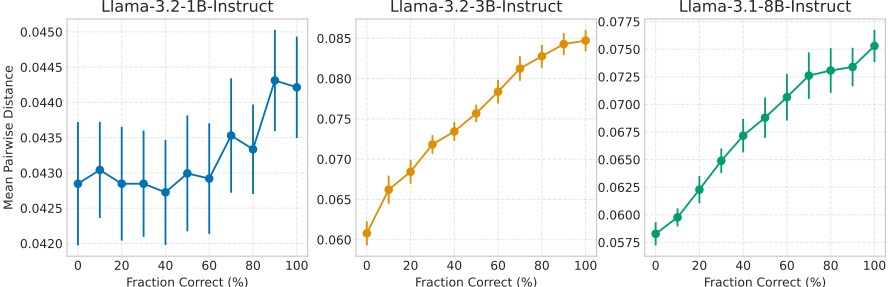

Figure 32: Downstream performance estimation on Multilingual MMLU (Thai) (containing results for Llama-3.2-1B-Instruct, Llama-3.2-3B-Instruct, and Llama-3.1-8B-Instruct).

### B.1.2   COMPARISON TO PERPLEXITY AS A HARDNESS SIGNAL

In §3.1 we used representation dispersion to rank slices of examples by their hardness for a fixed model and dataset. Here we replicate the same controlled slicing protocol but compare dispersion against the model's own perplexity on the query text as two competing label-free hardness signals. For each dataset (MMLU and ARC-CHALLENGE) and each model size (Llama-3.2-1B/3B-Instruct and Llama-3.1-8B-Instruct), we construct synthetic slices with target fractions of correct answers from 0% to 100% in steps of 10%. For every slice, we compute (i) the mean dispersion of the slice and (ii) the mean perplexity of the slice, averaging over 10 random seeds.

| Model | Frac. correct (%) | Perplexity | Dispersion |
|-------|-------------------|------------|------------|
| 1B | 100 | $5.273 \pm 0.063$ | $0.0584 \pm 0.0015$ |
|    | 90  | $5.232 \pm 0.056$ | $0.0578 \pm 0.0016$ |
|    | 80  | $5.196 \pm 0.068$ | $0.0559 \pm 0.0016$ |
|    | 70  | $5.201 \pm 0.064$ | $0.0553 \pm 0.0015$ |
|    | 60  | $5.178 \pm 0.050$ | $0.0528 \pm 0.0013$ |
|    | 50  | $5.166 \pm 0.033$ | $0.0513 \pm 0.0013$ |
|    | 40  | $5.147 \pm 0.044$ | $0.0504 \pm 0.0013$ |
|    | 30  | $5.092 \pm 0.054$ | $0.0488 \pm 0.0013$ |
|    | 20  | $5.038 \pm 0.042$ | $0.0470 \pm 0.0010$ |
|    | 10  | $5.070 \pm 0.032$ | $0.0450 \pm 0.0008$ |
|    | 0   | $5.058 \pm 0.033$ | $0.0425 \pm 0.0008$ |
| 3B | 100 | $4.740 \pm 0.055$ | $0.1525 \pm 0.0018$ |
|    | 90  | $4.753 \pm 0.043$ | $0.1507 \pm 0.0023$ |
|    | 80  | $4.761 \pm 0.044$ | $0.1468 \pm 0.0023$ |
|    | 70  | $4.738 \pm 0.040$ | $0.1414 \pm 0.0026$ |
|    | 60  | $4.730 \pm 0.041$ | $0.1355 \pm 0.0022$ |
|    | 50  | $4.760 \pm 0.050$ | $0.1298 \pm 0.0019$ |
|    | 40  | $4.803 \pm 0.049$ | $0.1221 \pm 0.0018$ |
|    | 30  | $4.780 \pm 0.042$ | $0.1158 \pm 0.0018$ |
|    | 20  | $4.742 \pm 0.033$ | $0.1077 \pm 0.0016$ |
|    | 10  | $4.762 \pm 0.040$ | $0.0987 \pm 0.0013$ |
|    | 0   | $4.751 \pm 0.025$ | $0.0923 \pm 0.0012$ |
| 8B | 100 | $4.528 \pm 0.044$ | $0.1395 \pm 0.0013$ |
|    | 90  | $4.500 \pm 0.043$ | $0.1374 \pm 0.0014$ |
|    | 80  | $4.477 \pm 0.047$ | $0.1342 \pm 0.0015$ |
|    | 70  | $4.472 \pm 0.047$ | $0.1310 \pm 0.0020$ |
|    | 60  | $4.488 \pm 0.048$ | $0.1244 \pm 0.0014$ |
|    | 50  | $4.498 \pm 0.051$ | $0.1186 \pm 0.0014$ |
|    | 40  | $4.488 \pm 0.045$ | $0.1127 \pm 0.0014$ |
|    | 30  | $4.458 \pm 0.049$ | $0.1048 \pm 0.0016$ |
|    | 20  | $4.438 \pm 0.045$ | $0.0979 \pm 0.0020$ |
|    | 10  | $4.419 \pm 0.044$ | $0.0894 \pm 0.0015$ |
|    | 0   | $4.418 \pm 0.043$ | $0.0801 \pm 0.0012$ |

Table 3: MMLU controlled slices: mean perplexity (ppl) and dispersion as a function of the fraction of correct answers. Dispersion is strongly monotone with correctness for all model sizes, whereas perplexity is nearly flat.

Across both datasets and all three model sizes, dispersion is strongly monotone with the fraction of correct answers, while perplexity is nearly constant and sometimes even slightly better (lower) on the hardest slices than on the easiest ones. This supports the claim that dispersion captures geometric information about contextual separation that is not reducible to next-token confidence alone.

| Model | Frac. correct (%) | Perplexity | Dispersion |
|---|---|---|---|
| 1B | 100 | $21.846 \pm 0.083$ | $0.0607 \pm 0.0013$ |
| | 90 | $21.670 \pm 0.105$ | $0.0589 \pm 0.0013$ |
| | 80 | $21.466 \pm 0.148$ | $0.0573 \pm 0.0012$ |
| | 70 | $21.289 \pm 0.130$ | $0.0553 \pm 0.0011$ |
| | 60 | $21.389 \pm 0.183$ | $0.0538 \pm 0.0010$ |
| | 50 | $21.094 \pm 0.226$ | $0.0516 \pm 0.0009$ |
| | 40 | $20.898 \pm 0.236$ | $0.0491 \pm 0.0008$ |
| | 30 | $20.842 \pm 0.225$ | $0.0456 \pm 0.0007$ |
| | 20 | $20.779 \pm 0.239$ | $0.0436 \pm 0.0007$ |
| | 10 | $20.561 \pm 0.237$ | $0.0426 \pm 0.0006$ |
| | 0 | $20.589 \pm 0.227$ | $0.0404 \pm 0.0007$ |
| 3B | 100 | $13.943 \pm 0.090$ | $0.1437 \pm 0.0020$ |
| | 90 | $13.917 \pm 0.099$ | $0.1418 \pm 0.0019$ |
| | 80 | $13.875 \pm 0.101$ | $0.1381 \pm 0.0022$ |
| | 70 | $13.957 \pm 0.111$ | $0.1328 \pm 0.0021$ |
| | 60 | $13.933 \pm 0.107$ | $0.1277 \pm 0.0024$ |
| | 50 | $14.052 \pm 0.084$ | $0.1232 \pm 0.0025$ |
| | 40 | $13.966 \pm 0.091$ | $0.1183 \pm 0.0018$ |
| | 30 | $13.939 \pm 0.120$ | $0.1111 \pm 0.0019$ |
| | 20 | $13.915 \pm 0.117$ | $0.1051 \pm 0.0020$ |
| | 10 | $13.862 \pm 0.081$ | $0.0986 \pm 0.0014$ |
| | 0 | $13.846 \pm 0.090$ | $0.0900 \pm 0.0010$ |
| 8B | 100 | $16.313 \pm 0.157$ | $0.1177 \pm 0.0014$ |
| | 90 | $16.272 \pm 0.160$ | $0.1157 \pm 0.0013$ |
| | 80 | $16.247 \pm 0.156$ | $0.1132 \pm 0.0013$ |
| | 70 | $16.102 \pm 0.189$ | $0.1105 \pm 0.0015$ |
| | 60 | $15.939 \pm 0.176$ | $0.1077 \pm 0.0013$ |
| | 50 | $16.052 \pm 0.158$ | $0.1045 \pm 0.0014$ |
| | 40 | $15.931 \pm 0.166$ | $0.0997 \pm 0.0011$ |
| | 30 | $15.869 \pm 0.109$ | $0.0951 \pm 0.0014$ |
| | 20 | $15.748 \pm 0.113$ | $0.0905 \pm 0.0012$ |
| | 10 | $15.755 \pm 0.147$ | $0.0844 \pm 0.0008$ |
| | 0 | $15.835 \pm 0.094$ | $0.0794 \pm 0.0007$ |

Table 4: ARC-CHALLENGE controlled slices: mean perplexity (ppl) and dispersion as a function of the fraction of correct answers. Again, dispersion is strongly monotone with correctness, whereas perplexity fails to separate easy from hard slices.

### B.1.3 Calibration by Dispersion

To test whether dispersion can be used as a calibrated proxy for accuracy on truly unseen data, we perform a "Calibration by Dispersion" experiment on MMLU. We partition the dataset into three disjoint subsets: (1) a *Calibration Set* (20% of the data), (2) a *Validation Set* (10%), and (3) a *Test Set* (70%). On the calibration set we use a dense sampling strategy to create synthetic batches with target accuracies ranging from 0% to 100% in 2% increments, which allows us to densely sample the shape of the Dispersion→Accuracy curve.

We then train a suite of regression models—linear, polynomial, isotonic, and random forest—to predict accuracy from dispersion, and select the model that minimizes prediction error on the validation set. Finally, we apply the selected regressor to the *unseen* test set (using dispersion only) and compare the predicted accuracy to the true accuracy. We repeat this procedure over 10 random seeds and report the mean absolute error (MAE) between predicted and true accuracy. Table 5 summarizes the results.

| Model | MAE (%) |
|---|---|
| Llama-3.2-1B-Instruct | 1.39 |
| Llama-3.2-3B-Instruct | 1.75 |
| Llama-3.1-8B-Instruct | 2.15 |

Table 5: Mean Absolute Error (MAE) between predicted accuracy (derived solely from dispersion) and true accuracy on a held-out 70% MMLU test set. Results are averaged over 10 random seeds. Dispersion supports highly accurate calibration, with absolute error of roughly 1.4–2.2 percentage points.

These results show that, for a fixed model and dataset, dispersion is not only monotonically related to correctness but can also be mapped to a well-calibrated accuracy estimate with small error. A practical workflow emerging from this experiment is:

1. Label a small random subsample of the data (e.g., 10–20%).

2. Fit a standard regression curve (e.g., isotonic or low-degree polynomial) from Dispersion → Accuracy on this subsample.

3. Apply the learned mapping to the remaining unlabeled data to estimate performance with a typical margin of error of about 2 percentage points.

While this calibration is, by construction, model- and domain-specific, it demonstrates that dispersion can serve as a practical, calibrated proxy for accuracy once a small labeled calibration set is available.

| Frac. correct (%) | 1B mean ± s.e. | 3B mean ± s.e. | 8B mean ± s.e. |
|---|---|---|---|
| 100 | 0.0350 ± 0.0004 | 0.0709 ± 0.0010 | 0.0750 ± 0.0012 |
| 90 | 0.0345 ± 0.0005 | 0.0694 ± 0.0009 | 0.0738 ± 0.0013 |
| 80 | 0.0343 ± 0.0004 | 0.0674 ± 0.0010 | 0.0714 ± 0.0012 |
| 70 | 0.0341 ± 0.0005 | 0.0665 ± 0.0009 | 0.0692 ± 0.0011 |
| 60 | 0.0335 ± 0.0006 | 0.0642 ± 0.0011 | 0.0669 ± 0.0009 |
| 50 | 0.0325 ± 0.0006 | 0.0620 ± 0.0011 | 0.0646 ± 0.0007 |
| 40 | 0.0319 ± 0.0006 | 0.0599 ± 0.0009 | 0.0627 ± 0.0006 |
| 30 | 0.0311 ± 0.0006 | 0.0583 ± 0.0010 | 0.0605 ± 0.0006 |
| 20 | 0.0306 ± 0.0006 | 0.0560 ± 0.0007 | 0.0578 ± 0.0009 |
| 10 | 0.0306 ± 0.0006 | 0.0540 ± 0.0006 | 0.0546 ± 0.0010 |
| 0 | 0.0302 ± 0.0005 | 0.0511 ± 0.0008 | 0.0513 ± 0.0008 |

Table 6: HELLASWAG_CHAT: mean dispersion (with standard error) as a function of the fraction of correct answers for Llama-3.2-1B/3B-Instruct and Llama-3.1-8B-Instruct.

| Frac. correct (%) | 1B mean ± s.e. | 3B mean ± s.e. | 8B mean ± s.e. |
|---|---|---|---|
| 100 | 0.7289 ± 0.0069 | 0.7266 ± 0.0064 | 0.7113 ± 0.0039 |
| 90 | 0.7277 ± 0.0068 | 0.7277 ± 0.0070 | 0.7141 ± 0.0043 |
| 80 | 0.7238 ± 0.0061 | 0.7250 ± 0.0048 | 0.7094 ± 0.0031 |
| 70 | 0.7203 ± 0.0054 | 0.7270 ± 0.0050 | 0.7074 ± 0.0031 |
| 60 | 0.7180 ± 0.0052 | 0.7246 ± 0.0035 | 0.7039 ± 0.0037 |
| 50 | 0.7145 ± 0.0053 | 0.7211 ± 0.0038 | 0.7043 ± 0.0055 |
| 40 | 0.7133 ± 0.0042 | 0.7238 ± 0.0033 | 0.7031 ± 0.0048 |
| 30 | 0.7113 ± 0.0045 | 0.7191 ± 0.0042 | 0.7031 ± 0.0036 |
| 20 | 0.7039 ± 0.0037 | 0.7156 ± 0.0056 | 0.6996 ± 0.0041 |
| 10 | 0.7070 ± 0.0030 | 0.7172 ± 0.0044 | 0.6961 ± 0.0021 |
| 0 | 0.7078 ± 0.0018 | 0.7164 ± 0.0039 | 0.6926 ± 0.0015 |

Table 7: IF_EVAL: mean dispersion (with standard error) as a function of the fraction of correct answers for Llama-3.2-1B/3B-Instruct and Llama-3.1-8B-Instruct.

### B.1.4 TESTING DISTRIBUTION SHIFT

To probe robustness under distribution shift, we further apply the controlled slicing experiment from §3.1 to three datasets with distributions very different from MMLU and ARC-CHALLENGE: HELLASWAG_CHAT, IF_EVAL, and QUAIL. For each dataset and for each of the three Llama-Instruct models, we construct slices with target fractions correct from 0% to 100% in steps of 10%, and compute the mean dispersion (with standard errors) for each slice. In all cases we observe the same qualitative trend: higher fractions correct correspond to larger mean dispersion, confirming that dispersion remains a reliable hardness signal even under substantial distribution shift. The full numbers are reported in Tables 6–8.

| Frac. correct (%) | 1B mean $\pm$ s.e. | 3B mean $\pm$ s.e. | 8B mean $\pm$ s.e. |
|---|---|---|---|
| 100 | $0.1938 \pm 0.0020$ | $0.1259 \pm 0.0024$ | $0.0893 \pm 0.0014$ |
| 90 | $0.1936 \pm 0.0020$ | $0.1234 \pm 0.0024$ | $0.0880 \pm 0.0015$ |
| 80 | $0.1941 \pm 0.0021$ | $0.1220 \pm 0.0023$ | $0.0865 \pm 0.0014$ |
| 70 | $0.1930 \pm 0.0021$ | $0.1207 \pm 0.0023$ | $0.0844 \pm 0.0014$ |
| 60 | $0.1931 \pm 0.0023$ | $0.1186 \pm 0.0028$ | $0.0826 \pm 0.0014$ |
| 50 | $0.1915 \pm 0.0021$ | $0.1165 \pm 0.0024$ | $0.0806 \pm 0.0012$ |
| 40 | $0.1913 \pm 0.0016$ | $0.1148 \pm 0.0018$ | $0.0788 \pm 0.0010$ |
| 30 | $0.1901 \pm 0.0013$ | $0.1122 \pm 0.0024$ | $0.0762 \pm 0.0007$ |
| 20 | $0.1898 \pm 0.0012$ | $0.1087 \pm 0.0022$ | $0.0735 \pm 0.0009$ |
| 10 | $0.1876 \pm 0.0011$ | $0.1046 \pm 0.0019$ | $0.0686 \pm 0.0009$ |
| 0 | $0.1884 \pm 0.0012$ | $0.1014 \pm 0.0020$ | $0.0656 \pm 0.0009$ |

Table 8: QUAIL: mean dispersion (with standard error) as a function of the fraction of correct answers for Llama-3.2-1B/3B-Instruct and Llama-3.1-8B-Instruct.

## B.2 Details Regarding Representation Dispersion for Model Selection

### B.2.1 Full Numeric Statistics

This appendix compiles the full numeric statistics that underpin the analyses in §3.2. We report complete Euclidean and cosine distance figures for every model variant, together with their task-specific accuracies, so that readers can perform fine-grained checks, reproduce our correlation calculations, and explore alternative dispersion metrics. Table 9, Table 10, Table 11 and Table 12 complement the visual summaries in Figure 9 by exposing each component of the *Dispersion Gap* in detail.

Table 9: **Embedding Dispersion vs. MATH Performance (Qwen Variants, Euclidean).** We show the mean ± standard error of **Euclidean** distances among digit embeddings (D–D), among non-math tokens (NM–NM), and between digits and non-math tokens (D–NM). We also list each model's accuracy on MATH (%).

| Model | Euclidean Distances | | | MATH (%) |
|---|---|---|---|---|
| | D–D | NM–NM | D–NM | |
| **Qwen2.5-1.5B** | $0.7006 \pm 0.0087$ | $1.4072 \pm 0.0268$ | $1.4331 \pm 0.0222$ | 35.0 |
| **Qwen2.5-Math-1.5B** | $0.8916 \pm 0.0111$ | $1.6423 \pm 0.0286$ | $1.6991 \pm 0.0203$ | 49.8 |
| **Distill-Qwen-1.5B** | $0.9406 \pm 0.0103$ | $1.6104 \pm 0.0247$ | $1.7014 \pm 0.0188$ | 83.9 |
| **Qwen2.5-7B** | $0.4505 \pm 0.0052$ | $0.8712 \pm 0.0189$ | $0.9840 \pm 0.0115$ | 49.8 |
| **Qwen2.5-Math-7B** | $0.6896 \pm 0.0076$ | $1.3479 \pm 0.0287$ | $1.4047 \pm 0.0205$ | 55.4 |
| **Distill-Qwen-7B** | $0.7216 \pm 0.0076$ | $1.3406 \pm 0.0292$ | $1.4244 \pm 0.0215$ | 92.8 |
| **Qwen2.5-14B** | $0.6993 \pm 0.0074$ | $1.5284 \pm 0.0314$ | $1.4550 \pm 0.0168$ | 55.6 |
| **Distill-Qwen-14B** | $0.7415 \pm 0.0073$ | $1.5223 \pm 0.0402$ | $1.4659 \pm 0.0229$ | 93.9 |

Table 10: **Embedding Dispersion vs. MATH Performance (Qwen Variants, Cosine).** We show the mean ± standard error of **Cosine** distances among digit embeddings (D–D), among non-math tokens (NM–NM), and between digits and non-math tokens (D–NM). We also list each model's accuracy on MATH (%).

| Model | Cosine Distances | | | MATH (%) |
|---|---|---|---|---|
| | D–D | NM–NM | D–NM | |
| **Qwen2.5-1.5B** | $0.2489 \pm 0.0060$ | $0.9334 \pm 0.0089$ | $1.0068 \pm 0.0217$ | 35.0 |
| **Qwen2.5-Math-1.5B** | $0.3347 \pm 0.0074$ | $0.8984 \pm 0.0110$ | $1.0755 \pm 0.0172$ | 49.8 |
| **Distill-Qwen-1.5B** | $0.3562 \pm 0.0073$ | $0.8973 \pm 0.0109$ | $1.0781 \pm 0.0165$ | 83.9 |
| **Qwen2.5-7B** | $0.1786 \pm 0.0042$ | $0.9360 \pm 0.0072$ | $1.0000 \pm 0.0159$ | 49.8 |
| **Qwen2.5-Math-7B** | $0.2672 \pm 0.0061$ | $0.9257 \pm 0.0089$ | $1.0554 \pm 0.0166$ | 55.4 |
| **Distill-Qwen-7B** | $0.2775 \pm 0.0061$ | $0.9260 \pm 0.0084$ | $1.0640 \pm 0.0177$ | 92.8 |
| **Qwen2.5-14B** | $0.2573 \pm 0.0053$ | $0.9333 \pm 0.0084$ | $0.9622 \pm 0.0100$ | 55.6 |
| **Distill-Qwen-14B** | $0.2811 \pm 0.0054$ | $0.9350 \pm 0.0111$ | $0.9701 \pm 0.0135$ | 93.9 |

Table 11: **Embedding Dispersion vs. HumanEval Performance (Llama2 vs. CodeLlama, Euclidean).** We report the mean ± standard error of **Euclidean** distances among code tokens (C–C), among non-code tokens (NC–NC), and between code and non-code tokens (C–NC). We also list each model's HumanEval pass@1 (%).

| Model | Euclidean Distances | | | HumanEval (%) |
|---|---|---|---|---|
| | C–C | NC–NC | C–NC | |
| **Llama2-7B** | $1.4961 \pm 0.0060$ | $1.3741 \pm 0.0313$ | $1.4465 \pm 0.0148$ | 12.2 |
| **CodeLlama-7B** | $2.3417 \pm 0.0110$ | $2.3437 \pm 0.0475$ | $2.3623 \pm 0.0249$ | 33.5 |
| **Llama2-13B** | $2.1274 \pm 0.0098$ | $1.9002 \pm 0.0524$ | $2.0324 \pm 0.0247$ | 20.1 |
| **CodeLlama-13B** | $2.5499 \pm 0.0134$ | $2.5305 \pm 0.0539$ | $2.5597 \pm 0.0275$ | 36.0 |

Table 12: **Embedding Dispersion vs. HumanEval Performance (Llama2 vs. CodeLlama, Cosine).** We report the mean $\pm$ standard error of **Cosine** distances among code tokens (C–C), among non-code tokens (NC–NC), and between code and non-code tokens (C–NC). We also list each model's HumanEval pass@1 (%).

| Model | Cosine Distances | | | HumanEval (%) |
|---|---|---|---|---|
| | C–C | NC–NC | C–NC | |
| **Llama2-7B** | $0.9603 \pm 0.0021$ | $0.9402 \pm 0.0075$ | $0.9652 \pm 0.0028$ | 12.2 |
| **CodeLlama-7B** | $0.9451 \pm 0.0027$ | $0.9486 \pm 0.0074$ | $0.9636 \pm 0.0037$ | 33.5 |
| **Llama2-13B** | $0.9340 \pm 0.0025$ | $0.9283 \pm 0.0101$ | $0.9460 \pm 0.0061$ | 20.1 |
| **CodeLlama-13B** | $0.9433 \pm 0.0027$ | $0.9435 \pm 0.0078$ | $0.9589 \pm 0.0037$ | 36.0 |

### B.2.2 ROBUSTNESS OF DISPERSION CORRELATION ACROSS TRAINING TRAJECTORIES

In §3.2, we demonstrated a strong correlation between the Dispersion Gap and model performance across different model families. To verify that this relationship is not an artifact of small sample sizes or specific model architectures, we conducted a high-resolution analysis of a single model's training trajectory.

**Setup.** We utilized the training checkpoints of the Olmo-7B model, sampling every 20,000 steps from step 120,000 to step 700,000. We exclude the initial 120,000 steps to bypass the 'burn-in' phase and focus on the stable training regime where representational geometry has consolidated. This yielded a set of $N = 30$ distinct checkpoints. For each checkpoint, we calculated the Dispersion Gap ($\mathcal{G}$) using our standard protocol and compared it against the model's zero-shot performance on GSM8K (Math) and HUMANEVAL (Code).

**Results.** As shown in Figure 33, we observe a remarkably strong monotonic relationship between dispersion and downstream accuracy throughout the training process.

- For Mathematical Reasoning, the correlation between the Dispersion Gap and GSM8K accuracy is $\rho = 0.90$ $(p < 10^{-11})$.
- For Code Generation, the correlation between the Dispersion Gap and HUMANEVAL pass@1 is $\rho = 0.95$ $(p < 10^{-15})$.

These results reinforce our claim that representation dispersion serves as a robust, label-free proxy for model capability. Notably, the dispersion metric tracks the underlying improvement of the model with high fidelity, maintaining a strong correlation even amidst the natural fluctuations inherent in generation-based evaluation benchmarks.

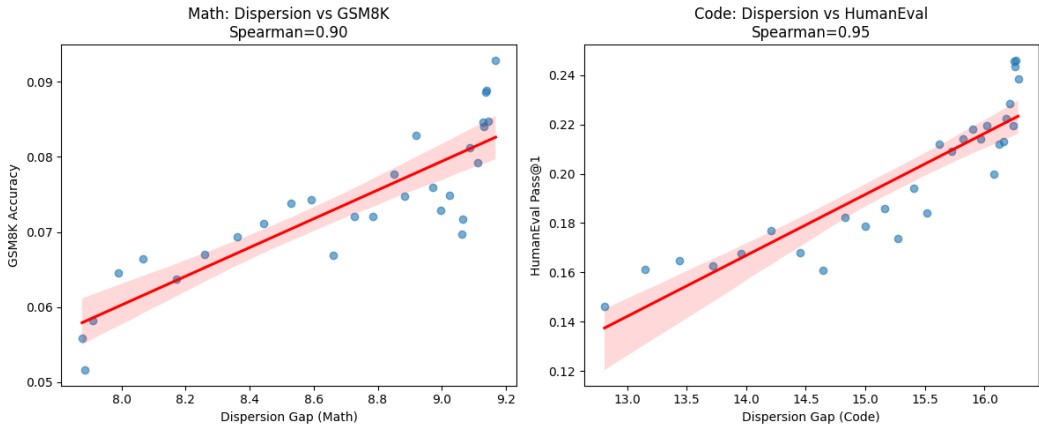

Figure 33: **Dispersion Gap tracks performance across the Olmo-7B training trajectory** ($N = 30$). We observe extremely high Spearman rank correlations ($\rho \approx 0.90$ for Math and $\rho \approx 0.95$ for Code) with high statistical significance ($p < 10^{-11}$). Shaded regions indicate the 95% confidence interval.

### B.3 DETAILS REGARDING LAYER SELECTION FOR kNN-LM

#### B.3.1 ADDITIONAL RESULTS

We present extended findings on sub-layer selection for *k*NN-LM. Figure 34 displays results for *four* GPT-2 variants (`distilgpt2`, `gpt2`, `gpt2-medium`, `gpt2-large`). As in the main text, each point represents a 512-token chunk of text, with its mean perplexity plotted against the sub-layer's average pairwise cosine distance (blue for the attention output, red for the feed-forward output). Interestingly, the negative correlation is weaker for the attention output than for the feed-forward output.

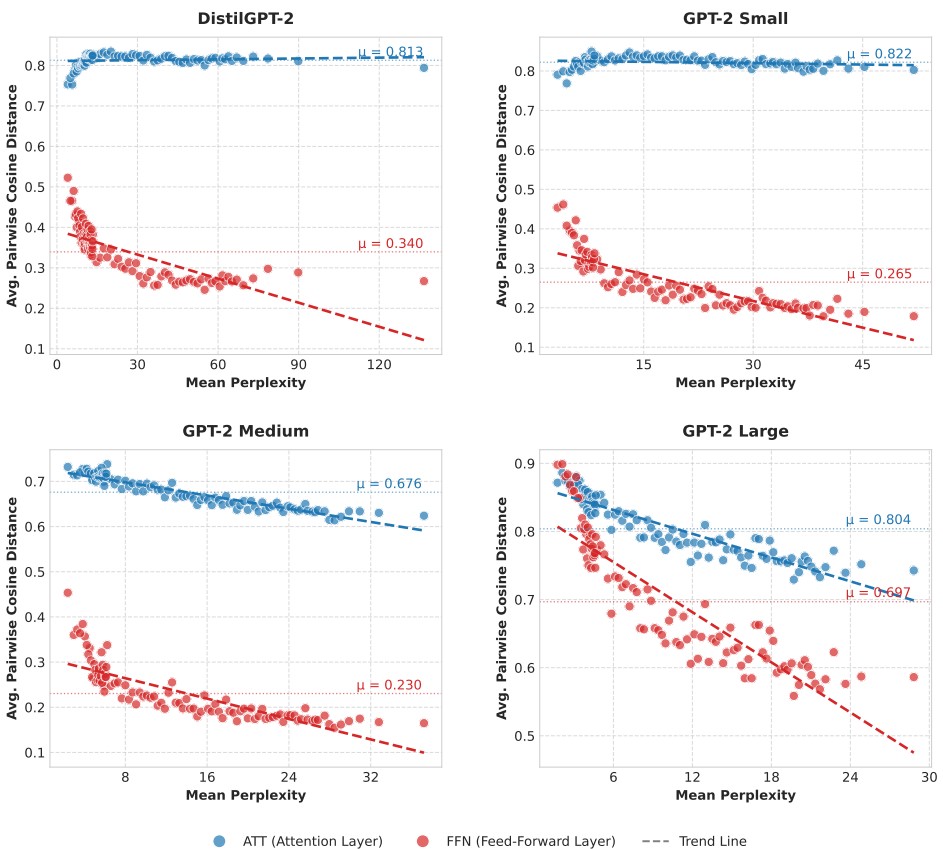

Figure 34: Mean perplexity vs. sub-layer average pairwise cosine distance for four GPT-2 variants (`distilgpt2`, `gpt2`, `gpt2-medium`, `gpt2-large`). Each point is a 512-token chunk of text.

## B.4 Details Regarding Incorporating Representation Dispersion

### B.4.1 Training Details

All experiments in §3.4 were conducted on NVIDIA A100 80GB GPUs.

**Single-Domain Setting.** We train GPT2-SMALL from scratch on WikiText, using a batch size of 64 and a block size (sequence length) of 512. The auxiliary loss weight $\lambda$ is tuned over the set:

$$\{0.5, 0.2, 0.1, 0.07, 0.05, 0.02, 0.01, 0.007, 0.005, 0.002, 0.001\}$$

We experiment with learning rates $\{1 \times 10^{-3}, 7 \times 10^{-4}, 5 \times 10^{-4}\}$.

**Cross-Domain Setting.** For joint WikiText + Python code training, we similarly use GPT2-SMALL (from scratch), a batch size of 128, and a block size of 256. The auxiliary loss weight $\lambda$ and learning rates are swept over the same sets as above:

- $\lambda$ values: $\{0.5, 0.2, 0.1, 0.07, 0.05, 0.02, 0.01, 0.007, 0.005, 0.002, 0.001\}$
- Learning rates: $\{1 \times 10^{-3}, 7 \times 10^{-4}, 5 \times 10^{-4}\}$

For both settings, we select $\lambda$ by validation for each learning rate. All experiments use standard AdamW optimizer settings unless otherwise specified.

### B.4.2 IMPACT OF ENFORCING CLUSTERIZATION ("SQUEEZE" ABLATION)

To further investigate the causal relationship between representation dispersion and model performance, we conducted an ablation study where we explicitly discouraged dispersion. In contrast to the "push-away" objective described in §3.4, here we added an auxiliary loss that minimizes the average pairwise cosine distance, effectively squeezing representations into tighter clusters.

We trained GPT2-SMALL on WIKITEXT-103 using the same hyperparameter sweep as our baseline experiments. Table 13 compares the test set perplexity of the Baseline model against the "Squeeze" model (where the auxiliary loss weight $\lambda = 0.1$).

| Learning Rate | Step | Perplexity (Lower is better) | | $\Delta$ PPL |
| --- | --- | --- | --- | --- |
| | | Baseline | Squeeze (Cluster) | |
| $1 \times 10^{-3}$ | 500 | 226.1 | 232.9 | +6.8 |
| | 1000 | 111.3 | 137.0 | +25.7 |
| $7 \times 10^{-4}$ | 500 | 195.0 | 206.1 | +11.1 |
| | 1000 | 96.7 | 124.2 | +27.5 |
| $5 \times 10^{-4}$ | 500 | 166.2 | 169.2 | +3.0 |
| | 1000 | 83.0 | 101.7 | +18.7 |

Table 13: **Effect of artificially reducing dispersion ("Squeeze" ablation).** Comparing the Baseline model to a model trained with an auxiliary loss that forces embeddings to cluster. Across all learning rates, restricting the geometry leads to significantly higher (worse) perplexity, reinforcing the causal link that adequate embedding breadth is necessary for strong predictive performance.

The results show a consistent and significant degradation in performance when dispersion is penalized. For the optimal learning rate ($5 \times 10^{-4}$), the perplexity worsens by over 18 points at step 1000. This substantial performance drop supports the hypothesis that representation dispersion is a functional driver of model performance; when the model is geometrically constrained from spreading its representations, its ability to minimize entropy and predict accurately is directly impaired.

### B.4.3 RELATION TO CONTRASTIVE AND REPULSIVE OBJECTIVES

Our auxiliary "spread-out" loss from §3.4 is conceptually related to prior contrastive and repulsive objectives that encourage more discriminative or isotropic representations, rather than being an entirely new idea. For example, Gunel et al. (2021) introduce a supervised contrastive loss for fine-tuning language models, in which representations of same-labeled examples are pulled together while representations of different classes are pushed apart. Likewise, Jain et al. (2023) propose ContraCLM, a contrastive framework for causal language models that explicitly improves the discrimination of token and sequence representations. Traditional contrastive losses (e.g., InfoNCE-style objectives) follow the same spirit: they add a term during training that repels embeddings from one another (except for designated positive pairs), mitigating representation collapse and improving downstream generalization.

Our formulation can be viewed as a deliberately simplified variant of these contrastive objectives. In contrast to InfoNCE-based losses, we do not define positive pairs or sample a specific set of negatives for each anchor. Instead, our loss considers *all* pairs of hidden states in the batch and directly maximizes their average cosine distance. This simplicity has two practical benefits: (i) it avoids sampling bias and the need for careful negative mining—every pair contributes equally to the repulsive force, so we are less sensitive to mini-batch composition or heuristic data augmentations; and (ii) it is easy to implement alongside the standard language-modeling loss, involving only a batch-wise cosine-distance computation without large softmax denominators or momentum-queue mechanisms. In this sense, our "push-away" loss can be viewed as a minimal, unsupervised contrastive regularizer that encourages a more uniform spread of representations on the unit sphere.

## C  LIMITATIONS

While our findings underscore a strong empirical link between representation dispersion and model performance, there are several limitations. First, our analyses focus on average pairwise cosine distances of final-layer representations, which may not capture all nuanced aspects of embedding geometry or model behavior. Second, although we observe consistent negative correlations between dispersion and perplexity across several model families and domains, causality cannot be definitively concluded; certain architectures or objectives may modulate this relationship in unforeseen ways. Third, our experiments center primarily on English text from standard benchmarks and a limited set of specialized domains (e.g. code, scientific abstracts). It remains unclear how well our observations extend to other languages, modalities, or highly domain-specific corpora. Further research is needed to fully understand these trade-offs and develop robust methods for controlling embedding geometry.

