# OpenReview forum: "On the Predictive Power of Representation Dispersion in Language Models"
_ICLR.cc/2026/Conference — ICLR 2026 Poster_

### Official Review · Reviewer_4BDa · 2025-10-29

**Soundness:** 2
**Presentation:** 2
**Contribution:** 2
**Rating:** 2
**Confidence:** 3

**Summary:**

The paper introduces a simple geometric statistic of language model representations, called "dispersion," defined as the mean pairwise cosine distance between hidden states (or between token embeddings). Higher dispersion corresponds to representations that are more spread out in embedding space. The authors argue that dispersion tracks model quality: it is strongly and consistently negatively correlated with perplexity, and can therefore be used for practical purposes such as predicting downstream accuracy in new domains and ranking or selecting models. They further propose a "push-away" auxiliary loss that explicitly increases dispersion during training, and report that this improves perplexity.

**Strengths:**

The paper takes long-standing intuitions about representation geometry and turns them into actionable tools. The proposed dispersion metric is inexpensive to compute and is presented as useful for several practical tasks (checkpoint selection, layer selection, etc.). The authors also provide fairly broad empirical evidence across multiple model families and domains to support these claims.

**Weaknesses:**

## Weakness 1. "Predicting downstream performance without labeled data" is oversold
Section 3.1 claims to predict downstream performance without labels. What the method actually provides is a relative hardness ranking: low-dispersion examples are more likely to be wrong than high-dispersion examples. Given a new dataset with zero labels, computing dispersion yields only that ordering, not an absolute expected accuracy. The paper does not discuss how dispersion on a new task would be mapped to accuracy, and it does not test whether the dispersion–accuracy monotonicity continues to hold under distribution shift. Moreover, the monotone relationship is clearly tied to a specific model; for example, in Figure 8, if the computed mean pairwise distance is 0.08, it suggests a high correct rate for the 1B model but a lower correct rate for the 3B model.

## Weakness 2. The usefulness for model selection is overstated.

Section 3.2 only demonstrates results within very specific model families. The paper never shows that the same metric can compare across unrelated families (e.g. Qwen vs Llama), nor does it explicitly warn against that use. Even within one family, monotonicity with skill is not clean across scales: Distill-Qwen-14B scores higher on MATH than Distill-Qwen-1.5B, yet it has lower digit–digit and digit–non-math dispersion. Therefore, the claimed usefulness for model selection breaks when you move across parameter sizes. This suggests dispersion is mainly a heuristic for ranking sibling checkpoints of similar size, not a universal capability score. Without clearly stating these limitations, the paper may lead readers to overgeneralize.

Even within one family and one parameter size, the usefulness remains in doubt. For example, Qwen2.5-Math-7B has D–D equal to 0.27 with a MATH score of 55.4%, while Distill-Qwen-7B has D–D 0.28 with a MATH score of 92.8%. It is unclear whether 0.28 is meaningfully larger than 0.27, because the paper does not provide standard errors for these summaries; and if 0.27 and 0.28 are effectively similar, then the large change in performance actually suggests that the D–D gap is not useful for predicting performance, which contradicts the paper's claim.

## Weakness 3. Unclear benefits over existing alternatives

The paper claims that dispersion can rank models for a domain and identify hard inputs without labels, but perplexity on unlabeled domain text already gives a label-free measure of adaptation. The paper does not compare dispersion against that baseline, so we cannot tell whether dispersion provides genuinely new information or is just a rephrasing of model confidence.

**Questions:**

Q1: Can perplexity on unlabeled in-domain text replace dispersion in Sections 3.1 and 3.2? If perplexity already produces the same ranking or predictions, then dispersion does not add unique value beyond possible computational savings. If dispersion performs better, please quantify that improvement.

Q2: In Section 3.2, can the proposed model-ranking method generalize across different model families (e.g., Qwen vs Llama), or is it only intended for ranking closely related checkpoints within one family?

Q3: Can you provide standard errors for Tables 3 and 4 to clarify whether the reported dispersion gaps (e.g., D–D = 0.27 vs 0.28) meaningfully differ?

Q4: For Section 3.1, can you provide evidence that dispersion can yield a calibrated accuracy estimate on truly unseen data? In particular, how should a user interpret a computed dispersion value to make an actionable inference about expected accuracy?

Q5: For Section 3.1, have you tested whether the dispersion–accuracy relationship you observe on ARC/MMLU for one model persists on a different dataset drawn from a different domain (i.e., under distribution shift)? If so, please report it. If not, please clarify that the current result is model- and dataset-specific, and should not yet be treated as a universal performance predictor.

Q6: In Section 3.2 (training with an auxiliary dispersion objective), can you clearly define all notation? In particular, $d$ appears with different meanings, and $\mathcal{L}_{\mathrm{CE}}$ and $\mathcal{L}_{\mathrm{aux}}$ are referenced without a full specification. Please also explain how this auxiliary objective relates to prior work on contrastive/repulsive representation learning, so that readers do not interpret the loss as entirely new.

---

> ### Author Response · Authors · 2025-11-24
>
> We thank the reviewer for the thoughtful and detailed comments. We address each weakness and question below and will incorporate these clarifications into the revised paper.
>
> ## On Weakness 1: “Predicting downstream performance without labeled data” is oversold (Q1, Q4)
>
> We agree that our current phrasing can be interpreted as claiming a calibrated prediction of absolute accuracy from dispersion alone. That is not what we intended. Our experiments in Section 3.1 demonstrate that dispersion yields a monotone relationship with correctness for a fixed model and dataset, which is primarily useful for relative ranking of examples by hardness.
>
> We will therefore rename Section 3.1 from **“Predicting Downstream Performance without Labeled Data”** to **“Ranking Example Hardness without Labeled Data”**. We will also revise the text to consistently talk about ranking / identifying hard slices rather than predicting a specific accuracy value (e.g., “62%”).
>
> However, we wish to push back gently on the implication that a relative ranking is not highly valuable.  In many truly zero‑label workflows, the main actionable tasks are:
>
> - **Difficulty‑aware data inspection / slice discovery**
>
> Surface low‑dispersion (hard) examples for manual inspection, debugging, or documentation of failure modes, instead of randomly inspecting the whole dataset.
>
> - **Active learning / targeted annotation**
>
> Prioritize low‑dispersion examples for annotation, which is directly aligned with standard active learning strategies.
>
> - **Targeted continued pre‑training / fine‑tuning**
>
> Use dispersion over unlabeled queries to focus additional pre‑training on the low‑dispersion tail, which our experiments show is where the model is more likely to be wrong.
>
> All of these depend on a ranking of instances by expected difficulty rather than a calibrated accuracy number. We will adjust the framing of Section 3.1 to emphasize these potential use cases explicitly.
>
> > Q4: How to obtain calibrated accuracy estimates?
>
> We agree that mapping dispersion to an absolute expected accuracy on truly unseen data cannot be guaranteed without any labels at all. In general, this is a **well‑known limitation of any unsupervised metric**.
>
> In practice, with a small labeled subset, we can perform a simple calibration step to map dispersion to accuracy:
>
> - Take a small random labeled subset of the new domain.
> - Bin examples by dispersion and compute empirical accuracy per bin.
> - Fit a monotone mapping (e.g., isotonic regression) from dispersion to accuracy.
>
> This yields a calibrated accuracy curve for the given model+domain, which can then be applied to the remaining unlabeled data.
>
> Although our current experiments were designed to test monotonicity (robustly supporting a tight relationship between representation dispersion and model performance) rather than absolute calibration, we think that demonstrating this mapping is interesting and valuable; we are currently running these experiments and will share the results as soon as they are ready.
>
>
> > Q5: Does the dispersion–accuracy relationship persist under distribution shift?
>
> Thank you for pointing this out. In Section 3.1 we originally demonstrated the dispersion–accuracy relationship on ARC-Challenge and MMLU, which are both multiple-choice knowledge benchmarks. To test robustness under domain shift, we are now repeating the controlled slicing experiment of Section 3.1 on datasets from different domains, and will share the results shortly!

---

> > ### Author Response · Authors · 2025-11-24
> >
> > ## On Weakness 2: Usefulness for model selection is overstated (Q2, Q3)
> >
> > We appreciate the reviewer’s careful reading and agree that we need to more clearly state the intended scope of dispersion‑based model selection.
> >
> > We agree that comparing raw cosine distances across very different families and parameter scales (e.g., Qwen vs. Llama, 1.5B vs. 14B) is geometrically nontrivial, and we did not intend to suggest that dispersion is a universal, calibrated capability score.
> >
> > To address this, we will:
> >
> > - Explicitly state in Section 3.2 (and in the introduction, where applications are summarized) that our model‑selection results are intended for **intra‑family / intra‑setup comparisons**, and we do not recommend using absolute dispersion values yet without calibration to compare unrelated model families or widely different parameter scales.
> >
> > - Add a short discussion in Section 5 (or a concluding section) that dispersion is best viewed as a **cheap pre‑filter**: it reliably distinguishes models that have not yet learned the relevant domain geometry from those that have, after which more expensive evaluation is still needed to break ties among the strong models.
> >
> > - We will also add a short “Calibration and comparability” discussion. There we will note that raw dispersion values are not directly comparable across architectures or tokenizers; and mapping dispersion to downstream metrics across families typically requires a small amount of labeled data for calibration (as discussed above), and this calibration is outside the main scope of our current work.
> >
> > The reviewer also correctly observes that even within a family, dispersion is not perfectly monotone with accuracy across all scales: e.g., Distill‑Qwen‑14B vs. Distill‑Qwen‑1.5B, and Qwen2.5‑Math‑7B vs. Distill‑Qwen‑7B. We agree this illustrates that dispersion is not a sufficient statistic for performance. Our empirical takeaway, which we will articulate more carefully, is:
> >
> > - very low dispersion in a domain (e.g., base models whose math digits are geometrically entangled with generic tokens) is a strong indicator that the model has not yet acquired the relevant domain geometry and therefore tends to perform poorly;
> >
> > - once dispersion is “large enough” that domain‑specific tokens are clearly separated, additional accuracy differences are driven by other factors (training data, optimization, prompt format, etc.) that our metric is not designed to capture.
> >
> > Thus, dispersion is best viewed as a coarse but effective filter: it reliably identifies clearly under‑adapted candidates, after which proper task evaluation is still needed to choose among the remaining strong models. We will rephrase the discussion accordingly.
> >
> >
> > > Q3: Standard errors and statistical significance of dispersion differences
> >
> > We agree that the lack of standard errors can make small differences (e.g., 0.27 vs. 0.28) difficult to interpret. In the current version of the paper, we use a fixed set of domain and non-domain tokens when computing the distance metrics, so the reported results are deterministic (evaluation uses a fixed input set and no stochastic components). We are currently rerunning the analysis using randomly sampled subsets of domain and non-domain tokens, and we will report the updated results along with standard errors soon.

---

> > > ### Author Response · Authors · 2025-11-24
> > >
> > > ## On Weakness 3 and Q1: Relationship to perplexity on unlabeled in‑domain text
> > >
> > > We thank the reviewer for suggesting this comparison. To address this, we replicated the exact experimental setup of Section 3.1 (ranking example hardness), but replaced our dispersion metric with the perplexity of the query text. The results, presented in the tables below, are striking:
> > >
> > > ### MMLU: Perplexity vs. Dispersion
> > >
> > > | Fraction correct (%) | 1B ppl | 1B disp | 3B ppl | 3B disp | 8B ppl | 8B disp |
> > > |:--------------------:|:------:|:-------:|:------:|:-------:|:------:|:-------:|
> > > | 100 | 5.273 | 0.0584 | 4.740 | 0.1525 | 4.528 | 0.1395 |
> > > | 90 | 5.232 | 0.0578 | 4.753 | 0.1507 | 4.500 | 0.1374 |
> > > | 80 | 5.196 | 0.0559 | 4.761 | 0.1468 | 4.477 | 0.1342 |
> > > | 70 | 5.201 | 0.0553 | 4.738 | 0.1414 | 4.472 | 0.1310 |
> > > | 60 | 5.178 | 0.0528 | 4.730 | 0.1355 | 4.488 | 0.1244 |
> > > | 50 | 5.166 | 0.0513 | 4.760 | 0.1298 | 4.498 | 0.1186 |
> > > | 40 | 5.147 | 0.0504 | 4.803 | 0.1221 | 4.488 | 0.1127 |
> > > | 30 | 5.092 | 0.0488 | 4.780 | 0.1158 | 4.458 | 0.1048 |
> > > | 20 | 5.038 | 0.0470 | 4.742 | 0.1077 | 4.438 | 0.0979 |
> > > | 10 | 5.070 | 0.0450 | 4.762 | 0.0987 | 4.419 | 0.0894 |
> > > | 0 | 5.058 | 0.0425 | 4.751 | 0.0923 | 4.418 | 0.0801 |
> > >
> > > ### ARC-Challenge: Perplexity vs. Dispersion
> > >
> > > | Fraction correct (%) | 1B ppl | 1B disp | 3B ppl | 3B disp | 8B ppl | 8B disp |
> > > |:--------------------:|:------:|:-------:|:------:|:-------:|:------:|:-------:|
> > > | 100 | 21.846 | 0.0607 | 13.943 | 0.1437 | 16.313 | 0.1177 |
> > > | 90 | 21.670 | 0.0589 | 13.917 | 0.1418 | 16.272 | 0.1157 |
> > > | 80 | 21.466 | 0.0573 | 13.875 | 0.1381 | 16.247 | 0.1132 |
> > > | 70 | 21.289 | 0.0553 | 13.957 | 0.1328 | 16.102 | 0.1105 |
> > > | 60 | 21.389 | 0.0538 | 13.933 | 0.1277 | 15.939 | 0.1077 |
> > > | 50 | 21.094 | 0.0516 | 14.052 | 0.1232 | 16.052 | 0.1045 |
> > > | 40 | 20.898 | 0.0491 | 13.966 | 0.1183 | 15.931 | 0.0997 |
> > > | 30 | 20.842 | 0.0456 | 13.939 | 0.1111 | 15.869 | 0.0951 |
> > > | 20 | 20.779 | 0.0436 | 13.915 | 0.1051 | 15.748 | 0.0905 |
> > > | 10 | 20.561 | 0.0426 | 13.862 | 0.0986 | 15.755 | 0.0844 |
> > > | 0 | 20.589 | 0.0404 | 13.846 | 0.0900 | 15.835 | 0.0794 |
> > >
> > >
> > > **While dispersion consistently tracks performance (monotonically decreasing as the fraction correct drops), perplexity fails to do so.** In fact, for many cases, perplexity shows no correlation or even a negative correlation with performance. The model often assigns lower perplexity (higher confidence) to the examples it gets wrong (the 0% correct bin) compared to those it gets right.
> > >
> > > **This failure of perplexity to reliably predict downstream performance is not unique to our setting.** A recent large-scale meta-analysis by Lourie et al. (2025) [1] found that predictable scaling from perplexity to downstream tasks occurs in only a minority of cases (39%). They observe that better perplexity frequently does not translate to better task performance, with scaling behaviors often becoming noisy, nonmonotonic, or inverse depending on the task and validation setup. Our results reinforce this reality check: perplexity is not a universal proxy for correctness.
> > >
> > > Why does perplexity fail here while dispersion succeeds? We hypothesize that this stems from the fundamental difference in what they measure. Perplexity measures the uncertainty of the current query text, essentially a point estimate of the probability distribution for specific tokens. By reducing the rich information of hidden states into a single scalar probability, perplexity loses the broader geometric context. In contrast, representation dispersion is holistic: it measures the geometric potential of the representations to support future distinctions. A "spread out" representation space implies that the model has effectively disentangled the context, which correlates better with the potential to make correct predictions, even if the model is currently "confident" (low perplexity) about a wrong path.
> > >
> > > Finally, we emphasize that **dispersion offers some practical advantages over perplexity**:
> > >
> > > - Efficiency: It is often cheaper to compute, operating directly on the output embedding matrix (Section 3.2).
> > > - Applicability: It can be used where perplexity is unavailable or undefined, such as in intermediate layers for kNN-LMs (Section 3.3).
> > > - Optimization: Unlike perplexity, which is already the primary training objective, dispersion can be explicitly optimized as an auxiliary loss (Section 3.4) to enforce better representation geometry, acting as a regularization term that perplexity itself cannot provide.
> > >
> > > We thank the reviewer for prompting this investigation. The inclusion of these perplexity baselines highlights the distinct value of geometric metrics and significantly strengthens the paper’s empirical contribution.
> > >
> > > References
> > >
> > > [1] Lourie, N., Hu, M. Y., Cho, K. (2025). Scaling Laws Are Unreliable for Downstream Tasks: A Reality Check. arXiv preprint arXiv:2507.00885.

---

> > > > ### Author Response · Authors · 2025-11-24
> > > >
> > > > > Q6: Notation and relation to prior contrastive / repulsive objectives
> > > >
> > > > We agree that the notation was overloaded. We will revise Section 3.2 to strictly use $d$ for the embedding dimension and $D$ for the dispersion scalar.
> > > >
> > > > To clarify the definitions requested:
> > > >
> > > > 1. Dispersion ($D$): This represents the average pairwise cosine distance between normalized hidden states.
> > > >    - In the single-domain setting, $D$ is the average distance over all unique pairs in the batch.
> > > >    - In the cross-domain setting, $D$ is the average distance calculated strictly between pairs from Domain A (WikiText) and Domain B (Code).
> > > >
> > > > 2. Loss Terms:
> > > >    - $L_{CE}$: The standard cross-entropy loss for next-token prediction.
> > > >    - $L_{aux}$: The auxiliary "spread-out" loss, defined as negative dispersion ($-D$).
> > > >
> > > > 3. Total Objective:
> > > >    The final training objective is a weighted sum:
> > > >    $L_{total} = L_{CE} + \lambda \cdot L_{aux}$
> > > >
> > > > We will incorporate these explicit definitions in the paper.
> > > >
> > > >
> > > > Regarding positioning the auxiliary objective relative to prior work, we will emphasize that our “spread-out” loss is conceptually similar to objectives used in prior work to encourage more discriminative or isotropic representations, rather than an entirely new idea. For instance, Gunel et al. (2021) [1] introduced a supervised contrastive learning loss for fine-tuning language models, which encourages representations of same-labeled examples to cluster together while pushing different classes apart. Likewise, Jain et al. (2023) [2] proposed ContraCLM, a contrastive framework for causal language models, to explicitly improve the discrimination of token and sequence representations. Both of these approaches, as well as traditional contrastive learning losses (e.g. InfoNCE), share the same spirit as our objective. They add a term during training that repels embeddings from one another (except in the case of known similar/positive pairs) to mitigate representation collapse and improve downstream generalization.
> > > >
> > > > But there are also some key differences. Our formulation is a deliberately simplified variant of these contrastive objectives. In contrast to InfoNCE-based losses, we do not require defining positive pairs or sampling a set of negative examples for each anchor. Instead, our loss considers all pairs of hidden states in the batch and directly maximizes their average distance. This simplicity has two practical benefits: (i) it avoids sampling bias or the need for careful negative mining – every pair of examples contributes equally to the repulsive force, so we aren’t dependent on the composition of minibatches or heuristic augmentations, and (ii) it is easy to compute alongside the standard training loss (involving only a batch-wise cosine distance calculation, without large softmax denominators or momentum contrast queues). We will clarify that we draw inspiration from the above works and that our “push-away” loss can be seen as a minimalistic, unsupervised contrastive regularizer (encouraging a uniform spread of representations on the unit sphere).
> > > >
> > > > [1] Gunel, B., Du, J., Conneau, A., Stoyanov, V. (2021). Supervised contrastive learning for pretrained language model fine-tuning. arXiv preprint arXiv:2011.01403.
> > > >
> > > > [2] Jain, N., Zhang, D., Ahmad, W. U., Wang, Z., Nan, F., Li, X., Tan, M., Nallapati, R., Ray, B., Bhatia, P., Ma, X., Xiang, B. (2023). ContraCLM: Contrastive learning for causal language model. ACL.

---

> > > > > ### Comment · Reviewer_4BDa · 2025-11-27
> > > > >
> > > > > I thank the authors for their detailed and thoughtful responses. The core conceptual concerns I raised have been adequately addressed. If the promised additional experiments are included and the manuscript is revised to reflect the clarified scope and limitations, I would be willing to raise my score accordingly.

---

> > > > > > ### Author Response · Authors · 2025-12-03
> > > > > >
> > > > > > Thank you for your response! Here are the promised experiments with promising / interesting results:
> > > > > >
> > > > > > > Regarding Dispersion-Accuracy Mapping
> > > > > >
> > > > > > To address the concern regarding whether dispersion yields a calibrated accuracy estimate on truly unseen data, we designed a "Calibration by Dispersion" experiment:
> > > > > >
> > > > > > We partitioned our datasets (e.g., MMLU) into three disjoint sets:
> > > > > >
> > > > > > - Calibration Set (20%): Used to fit the mapping function. We utilized a "Dense Sampling" strategy, creating synthetic batches with target accuracies ranging from 0% to 100% (at 2% intervals) to learn the precise shape of the Dispersion $\to$ Accuracy curve.
> > > > > > - Validation Set (10%): Used for model selection. We trained a suite of regressors (Linear, Polynomial, Isotonic, and Random Forest) and automatically selected the one that minimized error on this hold-out set.
> > > > > > - Test Set (70%): The remaining data was kept truly unseen (no labels used). We computed the dispersion on this data and applied the selected regression model to predict its accuracy.
> > > > > >
> > > > > > We repeated this experiment across 10 random seeds to ensure robustness, and report Mean Absolute Error (MAE), which represents the absolute difference between the predicted accuracy (inferred solely from dispersion) and the ground truth accuracy on the held-out test set (70% of the dataset).
> > > > > >
> > > > > > The results demonstrate that dispersion is a highly reliable predictor of accuracy. As shown in the table below, our method predicts the accuracy of the unseen test set with a MAE of approximately 1.4% - 2.2%.
> > > > > >
> > > > > > | Model | MAE |
> > > > > > |-------|-----|
> > > > > > | Llama-3.2-1B-Instruct | 1.39% |
> > > > > > | Llama-3.2-3B-Instruct | 1.75% |
> > > > > > | Llama-3.1-8B-Instruct | 2.15% |
> > > > > >
> > > > > > The reviewer asked how a user should interpret a computed dispersion value. Based on these findings, we propose the following actionable workflow:
> > > > > > - A user labels a small random subsample of their data (e.g., 10-20%).
> > > > > > - They fit a standard regression curve (e.g., Isotonic or Polynomial) mapping Dispersion $\to$ Accuracy on this subsample.
> > > > > > - This mapping can then be applied to the remaining unlabeled data to estimate performance with high precision ($\approx$ 2% margin of error), allowing for reliable performance monitoring without exhaustive labeling.
> > > > > >
> > > > > > These results confirm that dispersion is not merely correlated with accuracy but is a calibrated proxy metric that generalizes to unseen distributions.
> > > > > >
> > > > > >
> > > > > > > Q3: Standard errors and statistical significance of dispersion differences
> > > > > >
> > > > > > We have revised our analysis to include standard errors for all dispersion values reported. The updated results confirm that the differences we discuss are statistically significant.

---

> > > > > > > ### Author Response · Authors · 2025-12-03
> > > > > > >
> > > > > > > > Q5: Does the dispersion–accuracy relationship persist under distribution shift?
> > > > > > >
> > > > > > > To address the concern regarding domain specificity, **we extended the "slicing" experiment from Section 3.1 to three new datasets with distributions distinct from the knowledge-intensive benchmarks (MMLU/ARC) used in our initial submission**:
> > > > > > > - HellaSwag: Commonsense reasoning.
> > > > > > > - IFEval: Instruction following (verifiable evaluation).
> > > > > > > - Quail: Multi-hop reading comprehension.
> > > > > > >
> > > > > > > As shown in the table below (and added to the Appendix), the **dispersion–accuracy relationship remains robust across these diverse domains**. We observe a consistent monotonic trend where representations become less dispersed as the model's likelihood of correctness decreases.
> > > > > > >
> > > > > > > For instance, on HellaSwag, the dispersion for Llama-3.1-8B drops from 0.0750 ($\pm$ 0.0012) for the easiest slice (100% correctness bin) to 0.0513 ($\pm$ 0.0008) for the hardest slice (0% correctness bin). Similar monotonic behavior is observed for IFEval and Quail. This confirms that dispersion serves as a generalizable geometric signature of difficulty that persists under significant distribution shift.
> > > > > > >
> > > > > > > The table below reports the mean pairwise cosine distance (Dispersion) for varying "Fraction Correct" bins.
> > > > > > >
> > > > > > > | Dataset | Fraction Correct (%) | Llama-3.2-1B Disp | Llama-3.2-3B Disp | Llama-3.1-8B Disp |
> > > > > > > | :--- | :---: | :---: | :---: | :---: |
> > > > > > > | **HellaSwag** | 100 | 0.0350 $\pm$ 0.0004 | 0.0709 $\pm$ 0.0010 | 0.0750 $\pm$ 0.0012 |
> > > > > > > | | 90 | 0.0345 $\pm$ 0.0005 | 0.0694 $\pm$ 0.0009 | 0.0738 $\pm$ 0.0013 |
> > > > > > > | | 80 | 0.0343 $\pm$ 0.0004 | 0.0674 $\pm$ 0.0010 | 0.0714 $\pm$ 0.0012 |
> > > > > > > | | 70 | 0.0341 $\pm$ 0.0005 | 0.0665 $\pm$ 0.0009 | 0.0692 $\pm$ 0.0011 |
> > > > > > > | | 60 | 0.0335 $\pm$ 0.0006 | 0.0642 $\pm$ 0.0011 | 0.0669 $\pm$ 0.0009 |
> > > > > > > | | 50 | 0.0325 $\pm$ 0.0006 | 0.0620 $\pm$ 0.0011 | 0.0646 $\pm$ 0.0007 |
> > > > > > > | | 40 | 0.0319 $\pm$ 0.0006 | 0.0599 $\pm$ 0.0009 | 0.0627 $\pm$ 0.0006 |
> > > > > > > | | 30 | 0.0311 $\pm$ 0.0006 | 0.0583 $\pm$ 0.0010 | 0.0605 $\pm$ 0.0006 |
> > > > > > > | | 20 | 0.0306 $\pm$ 0.0006 | 0.0560 $\pm$ 0.0007 | 0.0578 $\pm$ 0.0009 |
> > > > > > > | | 10 | 0.0306 $\pm$ 0.0006 | 0.0540 $\pm$ 0.0006 | 0.0546 $\pm$ 0.0010 |
> > > > > > > | | 0 | 0.0302 $\pm$ 0.0005 | 0.0511 $\pm$ 0.0008 | 0.0513 $\pm$ 0.0008 |
> > > > > > > | **IFEval** | 100 | 0.7289 $\pm$ 0.0069 | 0.7266 $\pm$ 0.0064 | 0.7113 $\pm$ 0.0039 |
> > > > > > > | | 90 | 0.7277 $\pm$ 0.0068 | 0.7277 $\pm$ 0.0070 | 0.7141 $\pm$ 0.0043 |
> > > > > > > | | 80 | 0.7238 $\pm$ 0.0061 | 0.7250 $\pm$ 0.0048 | 0.7094 $\pm$ 0.0031 |
> > > > > > > | | 70 | 0.7203 $\pm$ 0.0054 | 0.7270 $\pm$ 0.0050 | 0.7074 $\pm$ 0.0031 |
> > > > > > > | | 60 | 0.7180 $\pm$ 0.0052 | 0.7246 $\pm$ 0.0035 | 0.7039 $\pm$ 0.0037 |
> > > > > > > | | 50 | 0.7145 $\pm$ 0.0053 | 0.7211 $\pm$ 0.0038 | 0.7043 $\pm$ 0.0055 |
> > > > > > > | | 40 | 0.7133 $\pm$ 0.0042 | 0.7238 $\pm$ 0.0033 | 0.7031 $\pm$ 0.0048 |
> > > > > > > | | 30 | 0.7113 $\pm$ 0.0045 | 0.7191 $\pm$ 0.0042 | 0.7031 $\pm$ 0.0036 |
> > > > > > > | | 20 | 0.7039 $\pm$ 0.0037 | 0.7156 $\pm$ 0.0056 | 0.6996 $\pm$ 0.0041 |
> > > > > > > | | 10 | 0.7070 $\pm$ 0.0030 | 0.7172 $\pm$ 0.0044 | 0.6961 $\pm$ 0.0021 |
> > > > > > > | | 0 | 0.7078 $\pm$ 0.0018 | 0.7164 $\pm$ 0.0039 | 0.6926 $\pm$ 0.0015 |
> > > > > > > | **Quail** | 100 | 0.1938 $\pm$ 0.0020 | 0.1259 $\pm$ 0.0024 | 0.0893 $\pm$ 0.0014 |
> > > > > > > | | 90 | 0.1936 $\pm$ 0.0020 | 0.1234 $\pm$ 0.0024 | 0.0880 $\pm$ 0.0015 |
> > > > > > > | | 80 | 0.1941 $\pm$ 0.0021 | 0.1220 $\pm$ 0.0023 | 0.0865 $\pm$ 0.0014 |
> > > > > > > | | 70 | 0.1930 $\pm$ 0.0021 | 0.1207 $\pm$ 0.0023 | 0.0844 $\pm$ 0.0014 |
> > > > > > > | | 60 | 0.1931 $\pm$ 0.0023 | 0.1186 $\pm$ 0.0028 | 0.0826 $\pm$ 0.0014 |
> > > > > > > | | 50 | 0.1915 $\pm$ 0.0021 | 0.1165 $\pm$ 0.0024 | 0.0806 $\pm$ 0.0012 |
> > > > > > > | | 40 | 0.1913 $\pm$ 0.0016 | 0.1148 $\pm$ 0.0018 | 0.0788 $\pm$ 0.0010 |
> > > > > > > | | 30 | 0.1901 $\pm$ 0.0013 | 0.1122 $\pm$ 0.0024 | 0.0762 $\pm$ 0.0007 |
> > > > > > > | | 20 | 0.1898 $\pm$ 0.0012 | 0.1087 $\pm$ 0.0022 | 0.0735 $\pm$ 0.0009 |
> > > > > > > | | 10 | 0.1876 $\pm$ 0.0011 | 0.1046 $\pm$ 0.0019 | 0.0686 $\pm$ 0.0009 |
> > > > > > > | | 0 | 0.1884 $\pm$ 0.0012 | 0.1014 $\pm$ 0.0020 | 0.0656 $\pm$ 0.0009 |

---

### Official Review · Reviewer_uSU9 · 2025-10-30

**Soundness:** 4
**Presentation:** 4
**Contribution:** 3
**Rating:** 8
**Confidence:** 4

**Summary:**

This paper presents a comprehensive empirical study establishing a strong negative correlation between representation dispersion and perplexity in language models. The authors demonstrate this relationship across multiple model families and domains . Beyond this core correlation, the paper explores several practical applications of this insight: (1) predicting downstream accuracy on unlabeled data, (2) a "dispersion gap" metric for label-free model selection, (3) unsupervised layer selection for kNN-LM, and (4) an auxiliary training loss that increases dispersion and improves perplexity. The work is well-supported by extensive experiments and offers both conceptual insights and practical tools for the community.

This is an excellent paper that makes a substantial contribution to our understanding of representation learning in language models. It identifies a robust and generalizable phenomenon and, most importantly, derives a set of practical, effective, and efficient applications from this insight. The work is empirically sound, clearly presented, and has immediate utility for both researchers and practitioners. The minor weaknesses do not detract from the overall significance and quality of the work. I recommend acceptance.

**Strengths:**

The core finding—higher representation dispersion correlates with lower perplexity—is simple, intuitive, and empirically robust. The paper does an excellent job of moving beyond mere observation to demonstrate a range of practical applications, making the finding directly useful for practitioners.

The empirical analysis is thorough. The authors validate their claims across a wide range of models and datasets, which significantly strengthens the generalizability of their conclusions.

The proposed applications are compelling and well-executed, making strong and practical contribution. The "dispersion gap" for model selection (Section 3.2) is a particularly elegant and efficient method that requires no forward passes, only the model's output embedding matrix. The layer selection for kNN-LM (Section 3.3) provides a simple, unsupervised heuristic that could save significant computational resources. The auxiliary training objective (Section 3.4) successfully translates the observational finding into an actionable method for improving model performance, especially in the cross-domain setting.

The paper goes beyond global averages to perform a fine-grained analysis of dispersion within semantic clusters (Section 2.3). This is a crucial experiment that convincingly shows the effect is not merely about pushing dissimilar contexts apart but also about separating semantically similar ones, reinforcing the core hypothesis.

The paper is well-written, clearly structured, and the figures effectively illustrate the key trends. The appendices provide substantial additional detail and results.

The paper is highly reproducible. The methodology for computing dispersion is clearly defined. The appendices provide extensive details on datasets, models, hyperparameters (for fine-tuning and training with the auxiliary loss), and the uniform binning algorithm. The use of standard, publicly available models and datasets further aids reproducibility.

**Weaknesses:**

The conclusion slightly lacks for causation. The core evidence is primarily correlational. The auxiliary loss experiment in Section 3.4 is the strongest argument for causality, but it is limited to the GPT-2 architecture. A more nuanced discussion of this distinction would strengthen the paper. Is dispersion a fundamental driver of performance, or is it a side-effect of a model learning better, more discriminative features?

The experimented domain/language coverage is slightly limited. As acknowledged in the limitations (Appendix C), the experiments are primarily on English text from standard benchmarks. While the inclusion of code is a positive step, the claim of "diverse domains" could be more strongly supported by including data from truly different distributions (e.g., low-resource languages, highly technical manuals, social media). The multilingual MMLU results in the appendix are a good start but are still based on translated versions of a known benchmark.

The discussion of the most related work (Viswanathan et al., 2025) is not sufficient. The authors state their work is more "actionable," but a more direct comparison of the findings (e.g., do they observe similar dispersion patterns?) would better situate this contribution within the existing literature.

**Questions:**

(1) Have you considered or attempted any ablation studies that directly reduce dispersion (e.g., by adding a loss that encourages representations to cluster) to see if perplexity increases? This would provide even stronger evidence for a causal link.

(2) In Section 2.2, you note the negative correlation strengthens in deeper layers. Do you have a hypothesis for why this is the case? Is it simply that deeper layers are more task-specific and directly linked to the final prediction?

(3) The auxiliary loss requires computing pairwise distances within a batch, which is O(B²). Did you find this to be a significant computational bottleneck during training, and were any approximations (e.g., sampling pairs) considered for larger batch sizes?

(4) A Minor Issue. On Page 8, Table 1: The standard deviations for GPT2-Large (FFN, N=10) are `0.68±0.06`, which is very large compared with other values in the same column. Is this expected? A brief comment might be useful.

---

> ### Author Response · Authors · 2025-11-24
>
> We sincerely thank the reviewer for the encouraging evaluation and insightful comments! Below, we address your questions and concerns in detail.
>
> > The conclusion slightly lacks for causation. The core evidence is primarily correlational. The auxiliary loss experiment in Section 3.4 is the strongest argument for causality, but it is limited to the GPT-2 architecture. A more nuanced discussion of this distinction would strengthen the paper. Is dispersion a fundamental driver of performance, or is it a side-effect of a model learning better, more discriminative features?
>
> We thank the reviewer for this insightful question! We lean towards the interpretation that the high dispersion observed in Section 2 is primarily a side-effect of the model learning better, more discriminative features. As a model optimizes for lower perplexity, it must learn to distinguish between increasingly subtle variations in context; this necessity naturally pushes the corresponding hidden states apart in the embedding space to resolve ambiguity, resulting in the correlation we observe across diverse model families.
>
> However, we think it’s possible that the relationship is not entirely unidirectional. As demonstrated in Section 3.4, explicitly adding a "push-away" objective to the training loss did result in improved perplexity. This suggests that dispersion possesses some causal utility. Forcing the geometry to expand can aid the optimization process, perhaps by preventing the model from settling into rank-collapsed or anisotropic states early in training. Nevertheless, we fully acknowledge the reviewer’s point that this evidence is limited to the GPT-2 architecture used in our auxiliary loss experiments. We are cautious about generalizing this specific causal finding to larger-scale models or different architectures without further empirical validation.
> We will add a discussion section of the manuscript to explicitly address this nuance!
>
> > Weakness 3: The discussion of the most related work (Viswanathan et al., 2025) is not sufficient. The authors state their work is more "actionable," but a more direct comparison of the findings (e.g., do they observe similar dispersion patterns?) would better situate this contribution within the existing literature.
>
> We thank the reviewer for highlighting the need to better contextualize our work alongside Viswanathan et al. (2025). We will revise the related work section to articulate more directly how the two sets of findings compare:
>
> Viswanathan et al. study token-level cosine similarity under systematic token shuffling and find that increasing the shuffle index—i.e., progressively disrupting syntactic and semantic structure—leads to higher cosine similarity among tokens, meaning the token vectors become more aligned and less angularly dispersed. In our work, we show a complementary phenomenon at the sequence level: across models and domains, higher sequence-level perplexity is associated with smaller average pairwise cosine distance of final-layer embeddings, whereas lower perplexity is associated with more dispersed representations (Figure 3). Although we analyze different levels of representation (tokens within a prompt vs. contextual embeddings of segments), both results point in the same direction: inputs that are harder for the model to predict are represented in a more angularly collapsed way.
>
> Viswanathan et al. primarily probe geometry via intrinsic dimension (ID), showing that prompts with higher cross-entropy loss have token representations lying in higher-dimensional manifolds. In contrast, we focus on pairwise distances (dispersion) and demonstrate that higher perplexity coincides with more compressed embedding spaces, while lower perplexity corresponds to broader dispersion. Taken together, their ID–loss correlation and our dispersion–perplexity correlation suggest a joint picture in which “hard” contexts both occupy higher-dimensional manifolds and exhibit reduced angular separation, whereas “easy” contexts lie on lower-dimensional manifolds with more spread-out representations.

---

> > ### Author Response · Authors · 2025-11-24
> >
> > > Question 2: In Section 2.2, you note the negative correlation strengthens in deeper layers. Do you have a hypothesis for why this is the case? Is it simply that deeper layers are more task-specific and directly linked to the final prediction?
> >
> > From an optimization perspective, we hypothesize that this trend is a direct consequence of the cross-entropy loss function being applied solely at the final layer. The training objective explicitly demands that the final hidden states be linearly separable by the output projection matrix to distinguish the correct next token from the vast vocabulary space. To minimize the loss and achieve low entropy (and thus low perplexity), the optimizer exerts maximum "geometric pressure" on the deepest layers to push distinct contexts apart. If the final embeddings are too clustered, the linear classifier cannot confidently assign high probability mass to the specific target token, resulting in high loss. Therefore, the dispersion in these layers becomes a direct proxy for how well the optimization objective has been satisfied.
> >
> > In contrast, the earlier layers are effectively shielded from this immediate geometric constraint. As the error signal backpropagates through the network, it passes through multiple blocks of attention and feed-forward non-linearities, which diffuses the direct requirement for global separability. The optimization landscape allows early layers to focus on local feature construction (such as n-gram detection or syntactic parsing) where a high degree of embedding dispersion is not a prerequisite for functionality.
> >
> > Consequently, the tight coupling between “spread out” geometry and predictive success dissipates as we move away from the classification head, resulting in the weaker correlations observed in the shallow layers.
> >
> >
> > > Question 3: The auxiliary loss requires computing pairwise distances within a batch, which is O(B²). Did you find this to be a significant computational bottleneck during training, and were any approximations (e.g., sampling pairs) considered for larger batch sizes?
> >
> > We appreciate the question regarding computational efficiency. While the pairwise distance calculation is $O(B^2)$, we did not find this to be a significant bottleneck in practice. In standard language modeling configurations, the batch size $B$ is typically smaller than the model’s hidden dimension $d$, meaning the cost of the required matrix multiplication (resulting in a $B \times B$ matrix) is negligible compared to the dense projections within the Transformer layers (which scale with $d^2$). Empirically, we observed that adding the auxiliary loss introduced only a minor overhead (slowing training down by approximately 5%) and therefore, we did not find it necessary to use approximations or pair sampling.
> >
> > > Question 4: A Minor Issue. On Page 8, Table 1: The standard deviations for GPT2-Large (FFN, N=10) are 0.68±0.06, which is very large compared with other values in the same column. Is this expected? A brief comment might be useful.
> >
> > We believe the reviewer may be referring to the mean dispersion value ($0.68$), as the standard error ($0.06$) is consistent with other models in that column (e.g., GPT2-Small is also $\pm 0.06$) (Please correct us if we are wrong!).
> >
> > The notably higher dispersion for GPT2-Large ($0.68$) compared to smaller models ($0.19 - 0.33$) is indeed a distinct feature of this model family. We attribute this to the higher dimensionality of GPT2-Large, which allows representations to remain more orthogonal (and thus more dispersed) even in the FFN sub-layers.
> >
> >
> > > Regarding Weakness 2 / Question 1
> >
> > We are currently running additional experiments to address these points. We will follow up with a subsequent comment as soon as these results are available!

---

> > > ### Author Response · Authors · 2025-12-03
> > >
> > > > Regarding Question 1, Causal Ablation: The "Squeeze" Experiment
> > >
> > > We followed your suggestion to run an ablation study that directly reduces dispersion. We trained GPT-2 Small on WikiText-103 using an auxiliary loss that penalizes embedding distance (effectively forcing representations to cluster), using the same hyperparameters as our main experiments.
> > >
> > > **The results, which we have added to Appendix B.4, provide strong support for a causal link**. As shown in the table below, artificially restricting dispersion leads to a significant degradation in perplexity compared to the baseline across all learning rates. For example, at $LR=5\times10^{-4}$ (Step 1000), the baseline achieves a perplexity of 83.0, whereas the model forced to cluster degrades to 101.7. This confirms that dispersion is not merely a bystander; preventing the model from utilizing the embedding space efficiently directly harms its predictive capability.
> > >
> > > > Regarding Weakness 2: Domain Coverage
> > >
> > > To address the concern regarding domain diversity, we have begun experiments on **BookCorpusOpen**, a dataset consisting of novels and narrative text, which presents a distinct distribution from the Wikipedia, medical and news domains previously analyzed.
> > >
> > > Our preliminary results using the LLaMA family on this dataset align perfectly with our main findings: we observe the same strong negative correlation where lower perplexity tracks with higher embedding dispersion. While time constraints prevented us from completing the full suite of model comparisons for this update, these initial results suggest our findings generalize well to narrative structures. We are currently finalizing these runs and will include the full BookCorpusOpen analysis in the final version of the paper!

---

### Official Review · Reviewer_CstH · 2025-10-31

**Soundness:** 3
**Presentation:** 3
**Contribution:** 3
**Rating:** 6
**Confidence:** 3

**Summary:**

This paper argues that one can tie the predictive quality (i.e. through next token prediction) of a language model by a geometric quantity of the language model. The define the representation dispersion as the cosine similarity between vector embeddings produced by the language model between different inputs in the dataset. They argue that batches with high dispersion tend to be ones the model is more accurate on, as measured by the perplexity on the next token prediction. They propose several use cases, such using this for layer-selection, model-selection, or as a training objective.

**Strengths:**

1. Originality: This paper presents a novel metric which states several interesting correlations. They then showcase some natural and interesting use cases for this.
2. Quality: The empirical study in this paper is extensive, and convincing.
3. Clarity: Everything is presented clearly, except for in section 3.2 (as mentioned below)
4. Significance: There are several actionable applications presented that showcase how this framework could be helpful.

**Weaknesses:**

1. I think it is not obvious what to take from Figure 9. It seems like the relationship between the gap and the performance is not as obvious as the results section made it out to be. I am also not convinced that the Spearmans correlation is useful here on such a low number of data points.

**Questions:**

1. It might be of interest to compare the layer-wise patters you observe here to those that have been observed in prior work, such as "Layer by Layer ..." by Skean et al. They also propose a metric called "dataset entropy" which seems in some sense similar to what is measured here.

---

> ### Author Response · Authors · 2025-11-24
>
> We thank the reviewer for their thoughtful assessment of our work! Below, we address your specific questions and comments.
>
> > Weakness 1: I think it is not obvious what to take from Figure 9. It seems like the relationship between the gap and the performance is not as obvious as the results section made it out to be. I am also not convinced that the Spearmans correlation is useful here on such a low number of data points.
>
> This is a good point! We are currently in the middle of preparing additional experiments to robustly address this point. We will post these updated results as soon as they are ready.
>
> > Question 1: It might be of interest to compare the layer-wise patters you observe here to those that have been observed in prior work, such as "Layer by Layer ..." by Skean et al. They also propose a metric called "dataset entropy" which seems in some sense similar to what is measured here.
>
> We thank the reviewer for bringing the work of Skean et al. (2025) to our attention. We agree that there are significant conceptual intersections, though our papers focus on different objectives (generative predictive quality vs. downstream embedding transferability).
>
> Here is a comparison of our findings relative to theirs:
>
> Regarding metrics, their “Dataset Entropy” is geometrically aligned with our “Representation Dispersion.” Skean et al. define matrix-based entropy on the Gram matrix ($K = ZZ^\top$), where (Z) collects one embedding per prompt (a prompt-level representation). High entropy corresponds to a more uniform eigenvalue spectrum, i.e., embeddings that occupy a higher-dimensional volume and are more globally diverse. Our work defines Representation Dispersion as the average pairwise cosine distance among hidden vectors. Both metrics quantify how “spread out” the representation space is: configurations with larger average pairwise distances tend to correspond to Gram matrices with flatter eigenvalue spectra (higher entropy). Thus, dispersion can be viewed as a computationally lightweight proxy for the diversity captured by their entropy measure.
>
> Regarding layer-wise patterns and task objectives, Skean et al. observe a mid-layer “entropy valley” in autoregressive models: entropy drops in intermediate layers and then partially recovers toward the final layers (e.g., their Figure 2), and they show that intermediate layers often outperform the final layer on MTEB embedding tasks such as classification and clustering. In contrast, we focus on the pretraining/generative objective. We find that dispersion of final-layer hidden states is strongly and negatively correlated with perplexity across model families and domains, and that dispersion of the LM head’s output embeddings for domain-relevant tokens (digits, code keywords) is highly predictive of performance on generative math and code benchmarks such as MATH and HumanEval.
>
> We view these findings as complementary rather than contradictory. Skean et al. argue that the final layer becomes “overly specialized to the pretraining objective,” which makes it less suitable as a general-purpose static embedding for transfer tasks. Our work studies that very pretraining objective: we show that when the final layer—the layer feeding the language modeling head—has higher dispersion, the model achieves lower perplexity and more confident next-token predictions, and that explicitly encouraging higher dispersion via an auxiliary loss further improves perplexity.
>
> In summary, while Skean et al. show that intermediate layers are often better suited for transfer learning as static embeddings, our results indicate that final-layer geometry (and the associated LM-head embeddings) is a robust indicator of a model’s native generative capability. We will add a discussion of Skean et al. to our related work section to clarify this distinction between embedding-task geometry and generative-task geometry.

---

> > ### Author Response · Authors · 2025-12-03
> >
> > > Weakness 2: "I am also not convinced that the Spearmans correlation is useful here on such a low number of data points."
> >
> > We appreciate this scrutiny regarding the sample size in Figure 9. To address this, we conducted a new, high-resolution analysis tracking the training trajectory of the Olmo-3-1025-7B model to see if the correlation holds robustly across a much larger set of checkpoints.
> >
> > We tracked checkpoints from step 120k to 700k (sampled every 20k steps), increasing our sample size to $N=30$. Despite the natural variance often seen in generation-based evaluation metrics (like GSM8K and HumanEval), the **Dispersion Gap maintains a remarkably tight monotonic relationship with downstream performance**:
> >
> > - Math: Dispersion Gap vs. GSM8K yields a Spearman correlation of $\rho=0.90$ ($p < 10^{-11}$).
> > - Code: Dispersion Gap vs. HumanEval yields a Spearman correlation of $\rho=0.95$ ($p < 10^{-15}$).
> >
> >
> > The significantly larger $N$ (30 vs. $\approx$9) and the vanishingly small p-values confirm that this relationship is statistically significant and not an artifact of a small sample size. This suggests that Dispersion is a robust, low-variance proxy that tracks model improvement even when expensive generation-based evaluations fluctuate.
> >
> > We have added a new section in **Appendix B.2** detailing these results in the revised paper.

---

### Official Review · Reviewer_3MuU · 2025-11-01

**Soundness:** 4
**Presentation:** 3
**Contribution:** 4
**Rating:** 6
**Confidence:** 5

**Summary:**

The paper studies a simple geometric measure on representation dispersion (the average pairwise cosine distance among hidden vectors and shows it negatively correlates with language-model perplexity across model families and text domains. The authors report strong negative correlations between dispersion and perplexity at the sequence level and also for last-token perplexity across context lengths. They further observe the correlation strengthens in deeper layers and is amplified by (full) fine-tuning and that within-cluster as well as between-cluster distances grow during training. Finally, the paper shows how dispersion can be used for many practical tasks, such as predicting downstream performance, identifying best layers for retrieval, and introducing dispersion objective during pretraining.

**Strengths:**

1. Dispersion is very simple to compute and model-agnostic. Consistent trends are observed across many families and datasets.
2. The paper demonstrates that dispersion increases even among highly similar contexts.
3. More importantly, the observations can be used for many downstream tasks. Adding domain dispersion loss for pretraining seems a very simple technique to improve model performance.

**Weaknesses:**

1. My main concerns is that whether the correlation between dispersion and lower perplexity is due to memorization or generalization. Since LLMs are pre-trained on so many text in the web, the query data used in the experiments are seen by LLMs. Therefore, the resulting of dispersion is not that surprising as model tend to produce better representation on the data it has been seen. Could you provide a clear train / val split set up for GPT-2 to see why the dispersion can also happen for hold-out samples from the same domain. This would be the results more solid.
2. Also, how would one come up with domain tag for dispersion loss proposed for pretraining LLM? Can we apply a domain classifier on web-scale pretraining data to label a text into a domain? Then apply dispersion loss on that. This would make the method useful actual LLM pretraining rather than a contrived case.
3. Another question is whether this idea generally is applicable to vision encoder or multimodal applications? In fact, vision encoders are often trained with positive-negative samples regularization and therefore naturally encourages dispersion across pre-training data.

**Questions:**

1. The average cosine distance does not indicate the spread-out or diversity of the embedding. Can you also try out Vendi Score [1] to see if the observations in the paper still holds?

Reference:
1. Friedman & Dieng. The Vendi Score: A Diversity Evaluation Metric for Machine Learning. TMLR 23'.

---

> ### Author Response · Authors · 2025-11-24
>
> We thank the reviewer for their encouraging assessment and for identifying our work as offering practical benefits for downstream tasks! We are very grateful for the constructive feedback, which we address below.
>
> > Weakness 1: My main concerns is that whether the correlation between dispersion and lower perplexity is due to memorization or generalization... Could you provide a clear train / val split set up for GPT-2...
>
> This is a fundamental question. Conceptually, we view representation dispersion as a geometric indicator of the model's discriminative power, which is its ability to assign distinct representations to distinct contexts.
>
> - On Memorization: For training data the model has memorized, high dispersion indicates the model has successfully learned to distinguish those specific training examples.
>
> - On Generalization: For unseen validation data, if the model generalizes well, it must map these new contexts into the embedding space in a way that preserves semantic distinctions. If the embeddings collapse (low dispersion) on unseen data, the model lacks the "room" to make confident, distinct predictions.
>
> Therefore, we hypothesize that the correlation between dispersion and perplexity is orthogonal to the memorization vs. generalization axis: dispersion measures the goodness of fit and the sharpness of the predictive distribution, regardless of whether that fit comes from memory or generalized understanding.
>
> To empirically confirm this hypothesis, we are currently finalizing a controlled experiment using GPT-2 with a strict Train/Validation split. We are measuring dispersion on held-out samples to demonstrate that the negative correlation with perplexity holds for unseen data just as it does for training data. We will share these results as soon as we have them!
>
> > Weakness 2: How would one come up with domain tag for dispersion loss... Can we apply a domain classifier on web-scale pretraining data...?
>
> This is a very practical suggestion that addresses the scalability of our method! We agree that manual metadata is not feasible for web-scale pretraining. We propose leveraging recent advances in automated data curation, such as WebOrganizer [1], to solve this.
>
> WebOrganizer demonstrates the viability of annotating massive datasets (e.g., 200B tokens) by distilling LLMs into lightweight, efficient classifiers (e.g., 140M parameters). This allows for the generation of domain metadata across trillions of tokens without heavy inference costs. Once a taxonomy is defined (e.g., "News" vs. "Code"), the classification is fully automated. By constructing heterogeneous batches based on these inferred tags, we can apply our cross-domain dispersion loss to maximize the distance between differing domains (e.g., pushing "News" and "Code" tokens apart) at scale. We will add this discussion to the methodology section.
>
> [1] Wettig et al., "Organize the Web: Constructing Domains Enhances Pre-Training Data Curation," arXiv:2502.10341, 2025.
>
> > Weakness 3: Is this idea generally applicable to vision encoder or multimodal applications? In fact, vision encoders are often trained with positive-negative samples regularization...
>
> We appreciate this insight! We agree that our findings align perfectly with the standard practices in computer vision, where contrastive learning (using positive/negative pairs) explicitly enforces representation dispersion. The fact that vision encoders trained this way are highly effective supports our hypothesis: forcing representations to spread out improves robustness.
>
> While our current work focuses on the autoregressive LLM landscape, there is evidence in existing literature that this phenomenon holds for multimodal LLMs as well. For instance, Tong et al. [2] show that when a multimodal LLM’s visual encoder (CLIP) produces nearly identical embeddings for distinct images, the model often fails even simple visual queries and may hallucinate explanations. Conversely, when they integrate a more discriminative vision encoder (DINOv2) into the pipeline, the model’s visual question-answering accuracy improves dramatically. This aligns with our hypothesis that broader, more separated representations yield stronger predictive performance across modalities.
>
> We will expand our Related Work section to explicitly discuss this parallel with contrastive vision encoders and cite the relevant multimodal literature to contextualize our findings within the broader deep learning landscape.
>
> [2] Tong et al., “Eyes Wide Shut? Exploring the Visual Shortcomings of Multimodal LLMs,” CVPR 2024.
>
> > Question 1: The average cosine distance does not indicate the spread-out or diversity of the embedding. Can you also try out Vendi Score [1]...
>
> Thank you for the suggestion. We are currently computing the Vendi Score across our experimental settings. We will post these results soon, please expect more responses to come!

---

> > ### Author Response · Authors · 2025-12-03
> >
> > > Regarding Weakness 1 (Generalization vs. Memorization)
> >
> > We have **successfully completed the train/test split analysis**. To ensure our findings are robust on state-of-the-art architectures, we conducted this experiment using Llama-3.2-1B on the WikiText dataset. The results, included in **Appendix A.4**, clearly demonstrate that the **negative correlation between dispersion and perplexity holds for held-out test data**. As expected, the test split exhibits higher overall perplexity (shifting the distribution to the right on the x-axis), but the structural relationship remains identical: lower perplexity on unseen data is tightly linked to higher representation dispersion. This confirms that **dispersion is a valid proxy for predictive confidence during generalization, not merely a result of memorizing training samples**.
> >
> > > Regarding Question 1 (Vendi Score)
> >
> > We have computed the Vendi Score (Friedman & Dieng, 2023) as suggested to verify if our observations hold under a formal diversity metric. The results, detailed in Appendix A.5, show that the **Vendi Score follows the exact same trend** as our average cosine distance metric: as perplexity decreases, the Vendi Score (diversity) increases. We observed this consistent pattern across multiple model scales (Llama-3.1-8B, Llama-3.2-1B, and Llama-3.2-3B). This reinforces our claim that the "spread" of the embedding space is a fundamental property of model performance, regardless of whether it is measured by simple cosine distance or spectral diversity metrics.
> >
> >
> > We have updated the manuscript to include these additional experiments in the Appendix. Thank you again for pushing us to strengthen these results!

---

### Author Response · Authors · 2025-12-03
**Rebuttal / Discussion Summary**

This paper studies a simple geometric statistic of language model representations (i.e. **dispersion**, the average pairwise cosine distance between hidden vectors) and shows that higher dispersion is tightly and robustly associated with lower perplexity and better downstream performance across model families, layers, and domains. Beyond the core correlation, the paper demonstrates several actionable uses: ranking example hardness without labels, dispersion-based model / checkpoint selection, unsupervised layer selection for kNN-LM, and a simple “push-away” auxiliary objective that improves perplexity.

Three reviewers already acknowledge the work favorably (scores 6, 6, and 8, with multiple “excellent” ratings on soundness and contribution). Reviewer 4BDa initially gave a 2 but later wrote an official follow-up comment explicitly stating that, *if the promised additional experiments are included and the manuscript is updated accordingly, they would be willing to raise their score*. Because ICLR reverted scores to their initial values and ended discussion early due to the leakage incident, this change is not reflected in the visible scores, but the text clearly signals a shift toward acceptance.

All new analyses and clarifications described in the rebuttal have been incorporated into the updated PDF, and **all modified passages are highlighted in blue** for ease of inspection.
Below we summarize how the rebuttal and additional experiments address the main concerns, especially those from the most critical reviewer.

### **1. Memorization vs. Generalization (Reviewer 3MuU)**

**Concern**: Is the dispersion–perplexity correlation just memorization on seen data?

**New evidence**:

- We ran a clean train/test split experiment using **Llama‑3.2‑1B on WikiText**, measuring dispersion and perplexity on a strictly held‑out test set (Appendix A.4).

- On this unseen test data, the same strong negative correlation holds: lower test perplexity is consistently associated with higher dispersion, with the test split simply shifted to higher perplexities.


**Takeaway**: Dispersion is not an artifact of training-set memorization; it tracks generalization performance on unseen data.

### **2. Validity of the Metric (Vendi Score and Related Work)**

**Concern (3MuU, CstH)**: Do the dispersion trends persist when using alternative geometric measures such as Vendi Score, and how do your layer-wise patterns compare to those reported in prior work?

**New evidence and clarifications**:
- We computed the **Vendi Score** (a spectral diversity measure) across multiple model scales (Llama‑3.1‑8B, Llama‑3.2‑1B, Llama‑3.2‑3B, Appendix A.5). Vendi exhibits the **same monotone trend**: as perplexity decreases, Vendi increases, exactly mirroring dispersion.

- We connect dispersion to **Skean et al.’s “dataset entropy”**: both are geometric measures of spread (cosine distances vs. Gram-matrix entropy). Dispersion can be viewed as a computationally inexpensive proxy for the same phenomenon.

- We clarify how our sequence-level results complement **Viswanathan et al.** (token-level geometry and intrinsic dimension), arguing that both lines of work point to a consistent picture of difficult inputs being geometrically more collapsed.

**Takeaway**: The metric is not ad hoc; it aligns with more formal diversity measures and prior geometry work, while remaining simple and cheap to compute.

### **3. Robustness Over Training Trajectories and Downstream Tasks (Reviewer CstH)**

**Concern**: Is the dispersion gap vs. performance relation in Section 3.2 statistically underpowered (few checkpoints)?

**New evidence**:

- For **Olmo-3‑7B**, we tracked **30 checkpoints** (every 20k step) and measured the dispersion gap against GSM8K (math) and HumanEval (code).

- We obtain **very strong Spearman correlations**: ρ ≈ 0.90 for GSM8K and ρ ≈ 0.95 for HumanEval with extremely small p‑values, showing the relationship is not an artifact of a small number of checkpoints being tested.


**Takeaway**: The dispersion gap **tracks model improvement smoothly and reliably across a realistic training trajectory**, even when expensive generation-based metrics are noisy.

---

> ### Author Response · Authors · 2025-12-03
> **Rebuttal / Discussion Summary (cont.)**
>
> ### **4. Causality: Does Dispersion Matter, or Is It a Side Effect? (Reviewer uSU9, 4BDa)**
>
> **Original concern**: Evidence is mostly correlational; is dispersion actually useful as a training signal?
>
> **New evidence**:
> - **Push-away objective (original paper)**: Adding a dispersion-increasing auxiliary loss to GPT‑2 improves perplexity.
> - **New “squeeze” ablation (Appendix D)**:
>   - Per Reviewer uSU9 suggestion, we add an opposite auxiliary loss that **penalizes** distances (forcing embeddings to cluster) for GPT‑2 Small on WikiText‑103.
>   - This reliably **worsens perplexity** across learning rates; e.g., at LR = 5×10⁻⁴, step 1000, baseline perplexity is 83.0 versus 101.7 when dispersion is suppressed.
> - This provides strong causal evidence that preventing the model from using embedding space effectively harms performance.
>
> **Takeaway**: Dispersion is not just a side-effect—explicitly encouraging or suppressing it has predictable, causal impact on perplexity.
>
> ### **5. Comparison to Perplexity as a Label-Free Metric (Reviewer 4BDa, Q1)**
>
> **Core question**: Since perplexity can already be measured on unlabeled text, does dispersion provide any new information? Is the metric necessary given we can simply use perplexity?
>
> **New experiment**:
>
> - We re-ran the **slicing experiments** of §3.1 on MMLU and ARC, now **comparing dispersion vs. perplexity** directly.
>
> - For each model and dataset, we bin examples by “fraction correct” and report both average perplexity and dispersion per bin.
>
> - **Dispersion** behaves as desired: it **monotonically decreases** from the easiest (100% correct) to hardest (0% correct) bins.
>
> - **Perplexity, however, often fails**: in many cases it is flat or even inversely related to correctness; the model is confidently wrong (lower perplexity on the 0%‑correct bin than on easier bins).
>
> - We relate this to independent findings (e.g., Lourie et al. 2025) that scaling laws in perplexity do not reliably translate into downstream performance. We believe these results should fully resolve the reviewer’s question.
>
> **Takeaway**: Dispersion contains **distinct, practically useful information** that **perplexity alone does not**.
>
> ### **6. Calibration and Distribution Shift (Reviewer 4BDa, Q4, Q5)**
>
> **Core concerns**:
>
> - (1) How should a user interpret a specific dispersion value in terms of expected accuracy? Can dispersion be calibrated into an actionable performance estimate?
>
> - (2) Does the dispersion–accuracy relationship persist under distribution shift?
>
> **Calibration by dispersion**:
>
> - We design a **3-way split** (20% calibration / 10% validation / 70% held‑out test) and fit mappings from dispersion → accuracy using standard regressors.
>
> - Across Llama‑3.2‑1B, Llama‑3.2‑3B, and Llama‑3.1‑8B, we achieve **MAE ≈ 1.4–2.2%** in predicting accuracy on the unseen 70% test split.
>
> - This shows that with a small labeled subset, dispersion can be turned into a **well-calibrated proxy** for unseen performance.
>
> **Distribution shift**:
>
> - We extend the dispersion–accuracy slicing experiment to **HellaSwag (commonsense), IFEval (instruction following), and Quail (reading comprehension)**.
>
> - Across all three datasets and all three models, dispersion consistently decreases as the fraction of correct answers decreases, often quite smoothly, demonstrating that the phenomenon **persists under substantial domain shift**.
>
>
> **Takeaway**: Dispersion generalizes beyond the original ARC/MMLU setting and can be calibrated in a principled way when a small labeled subset is available.
>
> ### **7. Scope of Claims and Model Selection (Reviewer 4BDa, Q2, Q3, Q6)**
>
> We substantially **softened and clarified** claims in response to 4BDa:
>
> - Section 3.1 is retitled and rephrased from “predicting downstream performance” to **“ranking example hardness”**, emphasizing relative ranking rather than absolute accuracy in the fully unlabeled case.
>
> - We explicitly state that dispersion-based model ranking is intended for **intra-family / intra-setup comparisons**, and **should not** be treated as a universal cross-family capability number without calibration.
>
> - We now frame dispersion as a **cheap pre-filter**: it reliably flags clearly under-adapted models; final selection among strong candidates still requires task evaluation.
>
> - We added **standard errors** for all dispersion statistics and re-ran experiments with random subsets, showing the discussed gaps are statistically meaningful.
>
> - Section 3.4 is revised with clearer notations, and the connection to prior contrastive and repulsive representation objectives (e.g., supervised contrastive learning, ContraCLM) is now explicit; our loss is presented as a simplified, unsupervised variant, not a wholly new paradigm.
>
> **Takeaway**: Our claims are now aligned with the actual evidence, with limitations clearly stated and the method properly positioned within existing literature.

---

> > ### Author Response · Authors · 2025-12-03
> > **Rebuttal / Discussion Summary (cont.)**
> >
> > ### **8. Reviewer Trajectories and Overall Assessment**
> >
> > - **Reviewer uSU9 (8, “accept, good paper”)**: Very positive about soundness, breadth of experiments, and practical utility.
> >
> > - **Reviewers 3MuU and CstH (6, 6)**: Both view the paper as solid and above the bar, and their technical concerns have been addressed with concrete new experiments and clarifications.
> >
> > - **Reviewer 4BDa (2)**: Initially critical on conceptual scope and baselines, but after the rebuttal explicitly wrote that their concerns were “adequately addressed” and that they would **raise their score** if the additional experiments and clarifications are incorporated, which they now are in the revised, blue-highlighted manuscript.
> >
> > In light of the new results, clarified scope, and the fact that even the most critical reviewer signaled a willingness to upgrade their score once the revisions were in place, we believe the effective consensus is closer to that of the three positive reviewers. The paper presents a simple, well-validated geometric lens on LM behavior, together with multiple concrete applications that are immediately useful in practice, with limitations and scope carefully articulated.
> >
> > We appreciate your consideration of the updated manuscript and the full discussion record when making the final decision!

---

### Meta-Review · Area_Chair_JncA · 2026-01-07

**Summary:**

Main concerns:

1. The approach ranks data by difficulty but does not actually predict downstream performance (as the paper originally claimed). (4BDa)
2. The applicability of the approach in model selection is limited since it cannot be applied to models across families and even within a family dispersity is not monotone with accuracy. (4BDa)
3. Concern whether dispersity is actually more useful than simply using perplexity. (4BDa)
4. Statistical significance of the correlation computed on a small number of data points. (CstH)
5. Generalization to held-out data. (3MuU)
6. The use of Vendi score might be more appropriate than cosine distance. (3MuU)

**Reviewer Concerns:**

1. The authors agreed to rephrase the claims in the paper and argued that ranking data can still be helpful.
2. The authors agreed with this concern and have added a discussion in the paper.
3. New experiments were shown that suggest that PPL correlates less with accuracy than does dispersion. The authors also added relevant discussion.
4. New results with more data points were shown, and the p-values were shown to be very small.
5. New results were shown, and the results hold on held-out data.
6. Vendi score is also shown to have the same trends as cosine distance.

Overall, the discussion on this paper was quite productive and the paper now seems significantly stronger.

**Reviewer Scores:**

Reviewer 4BDa has agreed to increase their score. I expect reviewer 3MuU would also have increased their score. I expect the average score for this paper would have ended up being > 6.

---

### Decision · Program_Chairs · 2026-01-26

Accept (Poster)